# Erythrocyte CD55 mediates the internalization of *Plasmodium falciparum* parasites

**Bikash Shakya[1], Saurabh D Patel[2], Yoshihiko Tani[3], Elizabeth S Egan[1]***

[1]Departments of Pediatrics and Microbiology & Immunology, Stanford University School of Medicine, Stanford, United States; [2]Zuckerman Institute, Columbia University, New York City, United States; [3]Japanese Red Cross Osaka Blood Center, Osaka, Japan

**Abstract** Invasion of human erythrocytes by the malaria parasite *Plasmodium falciparum* is a multi-step process. Previously, a forward genetic screen for *P. falciparum* host factors identified erythrocyte CD55 as essential for invasion, but its specific role and how it interfaces with the other factors that mediate this complex process are unknown. Using CRISPR-Cas9 editing, antibody-based inhibition, and live cell imaging, here we show that CD55 is specifically required for parasite internalization. Pre-invasion kinetics, erythrocyte deformability, and echinocytosis were not influenced by CD55, but entry was inhibited when CD55 was blocked or absent. Visualization of parasites attached to CD55-null erythrocytes points to a role for CD55 in stability and/or progression of the moving junction. Our findings demonstrate that CD55 acts after discharge of the parasite's rhoptry organelles, and plays a unique role relative to all other invasion receptors. As the requirement for CD55 is strain-transcendent, these results suggest that CD55 or its interacting partners may hold potential as therapeutic targets for malaria.

*For correspondence:
eegan@stanford.edu

Competing interests: The authors declare that no competing interests exist.

## Introduction

Malaria is caused by Apicomplexan parasites of the genus *Plasmodium*, of which *Plasmodium falciparum* is responsible for the majority of severe disease cases in humans. One of the world's major public health problems, malaria causes an estimated 216 million infections and ~445,000 deaths annually, primarily among young children and pregnant women (*WHO, 2018*). *P. falciparum* has a complex life cycle involving stages in the human and mosquito, but disease only occurs during the blood stage, when parasites infect and replicate in human red blood cells (RBCs). As *P. falciparum* is an obligate intracellular parasite, understanding the molecular determinants of its developmental cycle within RBCs may lead to new therapies. For example, a number of *Plasmodium* proteins that play key roles during erythrocyte invasion have shown promise as vaccine candidates (*Ord et al., 2015*; *Sack et al., 2017*; *Salinas et al., 2019*). Since natural genetic variation in red cells can influence innate susceptibility to malaria, host erythrocyte factors may also hold potential as therapeutic targets (*Taylor and Fairhurst, 2014*). The identification and study of such factors, however, has been severely limited by the intractability of mature RBCs, which lack a nucleus and DNA.

*P. falciparum* invasion of erythrocytes involves a series of coordinated events that unfold rapidly over the course of ~2 min. These events can be divided into three phases: pre-invasion, active invasion, and echinocytosis (*Gilson and Crabb, 2009*; *Weiss et al., 2015*). The process is initiated with the rupture of a 'mother' parasite, termed a schizont, which releases up to 32 daughter merozoites. During the pre-invasion phase, a free merozoite makes initial contact with the red cell, stimulating shallow deformation of the host cell plasma membrane. Next, the merozoite reorients so its apically-localized organelles are abutting the cell surface. Reorientation is associated with significant

membrane deformation and involves interactions between *P. falciparum* ligands such as the erythrocyte binding antigen (EBA) and reticulocyte binding-like homologues (Rh) family proteins, and receptors on the red cell surface (*Gilson and Crabb, 2009*; *Paul et al., 2015*; *Riglar et al., 2011*; *Tham et al., 2012*; *Weiss et al., 2015*). Several ligand and receptor pairs have been shown to act at this stage, often in a strain-specific manner (e.g. PfEBA-175 and GYPA; PfEBA-181 and GYPB PfEBA-140 and GYPC; and PfRh4 and CR1), but experimental data suggest their roles in apical reorientation and host cell deformation are largely functionally redundant (*Tham et al., 2012*).

The only receptor-ligand interaction known to be essential during the pre-invasion phase involves basigin and PfRH5, which exists in a complex with PfRipr and CyRPA (*Chen et al., 2011*; *Crosnier et al., 2011*; *Dreyer et al., 2012*; *Reddy et al., 2015*). Binding of the PfRH5 complex to basigin is required for discharge of the rhoptry organelles into the invaded cell and is associated with a calcium spike, potentially due to formation of a pore at the erythrocyte surface (*Volz et al., 2016*; *Weiss et al., 2015*). Blocking the interaction between PfRH5 and basigin with specific antibodies prevents invasion (*Crosnier et al., 2011*; *Patel et al., 2013*).

Discharge of the rhoptry organelles heralds the start of active invasion. Among the proteins injected from the rhoptries are those of the RON complex (RON 2, RON 4 and RON 5), which together form a receptor for binding by the PfAMA1 protein localized on the merozoite surface (*Alexander et al., 2006*; *Alexander et al., 2005*; *Richard et al., 2010*). The interaction between PfAMA1 and the RON complex forms a moving junction between the parasite and host cell that is believed to be entirely parasite-derived (*Besteiro et al., 2011*; *Besteiro et al., 2009*; *Harvey et al., 2014*; *Koch and Baum, 2016*). The moving junction provides an anchoring point for the merozoite to actively invade using its own actinomyosin motor; inhibiting the interaction between PfAMA1 and RON prevents invasion (*Richard et al., 2010*; *Srinivasan et al., 2011*; *Yap et al., 2014*).

As invasion proceeds, a parasitophorous vacuole is formed from components of the host cell membrane and rhoptries, yielding a protective niche for development of the new daughter parasite. The third phase, echinocytosis, is a transient period of cell dehydration and shrinkage observed after invasion; current evidence suggests it is triggered by discharge of the rhoptry contents, prior to and independent of active invasion. Echinocytosis is inhibited by reagents that prevent rhoptry discharge, such as antibodies targeting basigin or PfRH5 (*Weiss et al., 2015*).

Most host factors known to play a role in *P. falciparum* invasion have been identified based on their ability to bind to established invasion ligands or from studies of rare natural mutants (*Bei and Duraisingh, 2012*). Given the inherent intractability of mature RBCs, which lack a nucleus and DNA, the use of genetic approaches to identify and characterize malaria host factors presents a logistical challenge (*Egan, 2018*). Recently, an shRNA-based forward genetic screen using cultured red cells (cRBCs) derived ex-vivo from primary human hematopoietic stem/progenitor cells (HSPCs) identified erythrocyte CD55 (aka DAF) as a critical host factor for *P. falciparum* invasion (*Egan et al., 2015*). A 70 kD extracellular glycoprotein anchored to the red cell membrane by a glycosylphosphatidylinositol (GPI) linkage, CD55 is broadly distributed in different tissues and secretions, including blood cells (*Cooling, 2015*; *Storry et al., 2010*). On erythrocytes, CD55 acts as a complement regulatory protein to prevent complement-mediated damage. On epithelial cells, it has been shown to act as a receptor for Group B coxsackie virus and Dr+ *E. coli* (*Cooling, 2015*; *Coyne and Bergelson, 2006*).

We have shown that *P. falciparum* invasion efficiency was reduced by ~50% in CD55-knockdown cRBCs, and natural CD55-null erythrocytes from two rare donors with the Inab phenotype were resistant to invasion (*Egan et al., 2015*). Importantly, the requirement for CD55 was strain-transcendent, suggesting that it plays a conserved role in *P. falciparum* invasion. However, the precise function of CD55 during invasion and how it may interface with established ligands or receptors is unknown.

In this study, we investigated the functional role of CD55 during *P. falciparum* invasion using CRISPR-Cas9 genome editing, antibody-based inhibition and live cell microscopy. We found that CD55 plays a critical role during *P. falciparum* invasion of mature erythrocytes, where it is specifically required for parasite internalization. As a host factor that acts after discharge of the parasite's rhoptry contents, CD55 plays a unique role relative to other receptors required for invasion, providing a crucial link between the ligand-receptor interactions important for adhesion and deformation and effective internalization.

## Results

### Generation of CD55-null red blood cells using CRISPR-Cas9

Previously, we observed that *P. falciparum* merozoites from several laboratory-adapted strains and clinical isolates displayed impaired invasion into cryopreserved CD55-null RBCs from two rare patients with the Inab phenotype (*Egan et al., 2015*). To study the requirement for CD55 in an isogenic background, we sought to generate CD55-null cRBCs from HSPCs using CRISPR-Cas9 genome editing by co-delivering single guide RNAs (sgRNAs) and Cas9 in a ribonucleoprotein complex (RNP), a method that has been used previously to generate null mutants in primary human CD34 + cells (*Hendel et al., 2015*). We designed two chemically modified sgRNAs targeting the 5' end of *CD55*, and delivered them individually or together into primary human CD34$^+$ HSPCs via nucleofection, in complex with recombinant Cas9 (*Figure 1A*). As a control, isogenic CD34$^+$ HSPCs from the same donor were nucleofected with Cas9 alone. After inducing the cells to differentiate down the erythroid lineage for 18 days, we observed a high percentage of CD55-null cRBCs in the RNP-transfected populations; a single sgRNA targeting exon one resulted in ~70% CD55-null cRBCs, while the combination of two sgRNAs targeting exons 1 and 2 increased the knockout efficiency to ~90% (*Figure 1B*). The cells proliferated ~10,000 fold and the enucleation rate was >90% in both the CD55-null cells and wild-type controls, demonstrating that the progenitors differentiated efficiently and that CD55 is not required for this process (*Figure 1—figure supplement 1*). The absence of CD55 expression in the mutant cRBCs was also confirmed using immunofluorescence assays (*Figure 1—figure supplement 2*).

### CD55 is required for *P. falciparum* invasion

To specifically assess the requirement for CD55 for *P. falciparum* invasion, we performed invasion assays using strain 3D7 in CD55-CRISPR cRBCs or in isogenic control cRBCs that had been differentiated for 18 days. While invasion into the CD55-CRISPR cRBCs was impaired relative to the control cells, it was only reduced by ~40% (*Figure 1C*). This subtle phenotype was reminiscent of the results for CD55-knockdown cRBCs, and differed from the strong invasion phenotype observed in erythrocytes from two CD55-null patients (*Egan et al., 2015*). Since cRBCs are less mature than erythrocytes from peripheral blood, we hypothesized that the relatively mild invasion phenotype observed in the CD55-null cRBCs may be explained by the overall maturation state of the ex vivo cultures on day 18.

To begin to determine whether cell maturity modifies the requirement for CD55 during *P. falciparum* invasion, we assessed the surface expression of CD71 in the cRBCs, which is highly expressed on erythroblasts and young reticulocytes, but quickly disappears as reticulocytes mature into erythrocytes (*Hu et al., 2013*). Flow cytometry on day 17 demonstrated that approximately 50% of the enucleated cRBCs were CD71-positive and ~50% were CD71 negative, indicating a mix of cell maturity in the population (*Figure 1D*). This was further validated using a reticulocyte stain (*Figure 1—figure supplement 3*), which showed that day 17 cells were ~50% reticulocytes and 50% erythrocytes. A timecourse over seven additional days of terminal differentiation revealed a progressive loss of CD71 expression, consistent with a change in the population structure to one dominated by erythrocytes rather than reticulocytes (*Figure 1D*).

To isolate a more homogeneous population of mature cRBCs for invasion assays, we differentiated the cRBCs for 21–22 days and then used anti-CD71 antibody-immobilized magnetic beads to deplete CD71-positive cells and enrich for CD71-negative cells (*Figure 1—figure supplement 4*). Using these highly mature cells, invasion assays with *P. falciparum* strain 3D7 showed a ~75% reduction in invasion into the CD55-CRISPR cRBCs as compared to isogenic control cRBCs in three biological replicate experiments (*Figure 1E*). Since our CRISPR-Cas9 genome editing strategy yielded a mixed population of ~90% CD55-null and 10% CD55-positive cells, we suspected that the residual invasion observed in the 'CD55-CRISPR' population was attributable to wild-type cells. This was confirmed by immunofluorescence assays showing that *P. falciparum* invasion was restricted to the cells expressing CD55: the parasitemia in the minority, CD55-positive cells in the population was 22.5%, whereas less than 0.5% of the CD55-null cRBCs were infected (*Figure 1F–G*). These results demonstrate that CD55 is critical for *P. falciparum* invasion of fully differentiated cRBCs, mimicking the prior observations with natural CD55-null erythrocytes, and validating CD55 as an essential host factor for

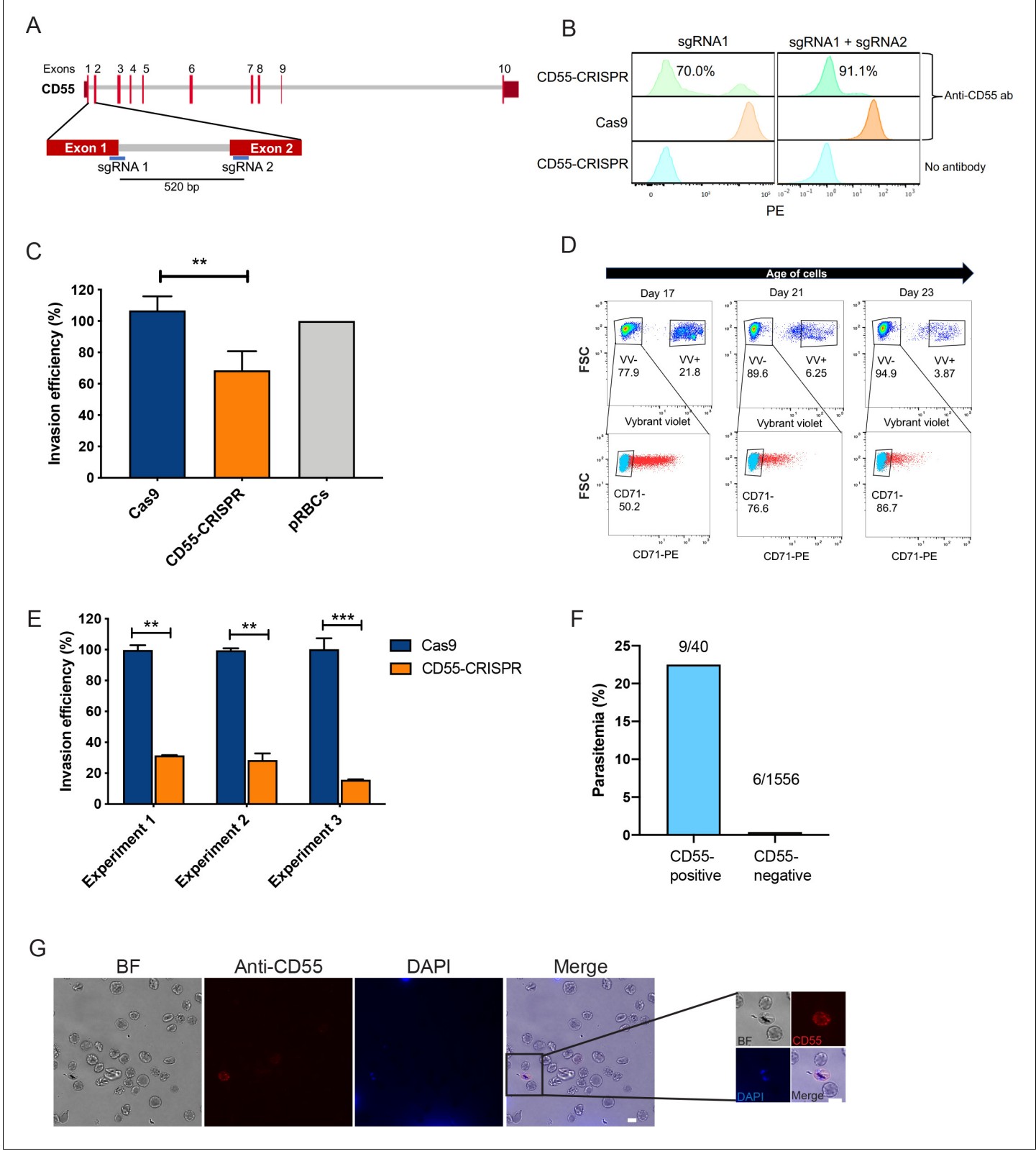

**Figure 1.** CD55 is required for *P. falciparum* invasion. (**A**) Schematic of *CD55* gene structure showing targeting sites of two single guide RNAs (sgRNAs). Vertical red lines indicate the positions of *CD55* exons. (**B**) Expression of CD55 on mutant (CD55-CRISPR) cRBCs generated with one sgRNA (left) or two sgRNAs (right), as compared to control (Cas9) cRBCs. (**C**) Invasion efficiency of *P. falciparum* 3D7 in Day 18 CD55-CRISPR cRBCs compared to isogenic controls (Cas9), relative to the invasion efficiency in peripheral blood erythrocytes (N = 4 biological replicates; n = 3 technical replicates; *Figure 1 continued on next page*

*Figure 1 continued*

error bars indicate SEM; **p<0.01). CD55-null cRBCs were generated using dual sgRNAs: sgRNA1, CD55-Cr1 and sgRNA2, CD55-Cr8. (D) Time course of expression of CD71 on cRBCs harvested on different days of differentiation. Enucleated versus nucleated cells were gated using a nuclear dye. (E) Invasion efficiency of *P. falciparum* strain 3D7 in CD71-negative, CD55-CRISPR cRBCs compared to CD71-negative isogenic controls (Cas9). Invasion efficiency is presented relative to the mean of Cas9 control. Three independent biological replicates are shown; error bars represent SEM (n = 2 or three technical replicates; **p<0.01, ***p<0.005). CD55-null cRBCs were generated using dual sgRNAs. (F) Parasitemia in population of CD55-CRISPR cRBCs, in which ~ 90% of the cells lack CD55 (CD55-negative), and the remaining are CD55-positive, as quantified by immunofluorescence assays. (G) Representative images of CD55-CRISPR cRBCs, including one *P. falciparum*-infected cell (magnified). Images are brightfield with fluorescence overlaid. Blue, dapi. Red, anti-CD55-PE. Scale bars indicate 10 μm.

The online version of this article includes the following source data and figure supplement(s) for figure 1:

**Source data 1.** Source data for invasion assays.
**Figure supplement 1.** Deletion of CD55 does not impact growth or maturation of cRBCs derived from CD34 +primary human HSPCs.
**Figure supplement 1—source data 1.** Source data for growth curves.
**Figure supplement 2.** Expression of CD55 on cRBCs.
**Figure supplement 3.** Reticulocyte staining of day 17 cRBCs.
**Figure supplement 4.** Enrichment of CD71- cRBCs by immunolabeled magnetic beads.

*P. falciparum* invasion. Based on previous observations that the expression levels of many RBC surface proteins decline during reticulocyte maturation (*Chu et al., 2018*; *Malleret et al., 2013*), we hypothesize that high levels of other receptors important for invasion, such as basigin, may explain the reduced requirement for CD55 for *P. falciparum* invasion of younger, CD71-positive reticulocytes.

## Antibodies targeting CD55 inhibit *P. falciparum* invasion

To determine if *P. falciparum* invasion can be inhibited using antibodies targeting CD55, we performed in vitro growth assays in the presence of three established anti-CD55 monoclonal antibodies individually and in combination. The antibodies each recognize a different short consensus repeat (SCR) in the CD55 ectodomain: BRIC 230, BRIC 110 and BRIC 216 target SCR1, SCR2 and SCR3, respectively. While none of the antibodies individually affected parasite growth, we observed a dose-dependent inhibition of parasite growth over 72 hr in the presence of all three anti-CD55 monoclonal antibodies combined, in comparison to an isotype control (*Figure 2A*). At the highest antibody concentration tested (500 μg/ml), the relative parasitemia was reduced by ~40% in the presence of the combined monoclonals, compared to the isotype control.

To further study anti-CD55-mediated inhibition of *P. falciparum* invasion, we generated a rabbit polyclonal antibody raised against the entire ectodomain of human erythrocyte CD55. Flow cytometry analysis confirmed that the purified IgG antibody recognizes an antigen on wild-type (WT) RBCs but not on CD55-null RBCs from an Inab donor (*Takahashi et al., 2008*), confirming its specificity for CD55 (*Figure 2—figure supplement 1*). In *P. falciparum* growth assays, we observed a dose-dependent inhibition of parasite growth in the presence of anti-CD55 antibody relative to isotype control, with a ~ 40% reduction in relative parasitemia at the highest concentration of antibody (400 μg/ml) (*Figure 2B*). This degree of inhibition was very similar to that seen for the pooled monoclonal antibodies; together, these findings suggest that blocking CD55 on the RBC can inhibit *P. falciparum* invasion.

We next tested the effect of anti-CD55 antibody on sialic acid-independent invasion pathways by treating cells with neuraminidase (NM) to remove sialic acid. As has been described previously, the growth of *P. falciparum* strain 3D7 was inhibited ~40% in NM-treated RBCs as compared to untreated cells in the absence of antibody (*Figure 2C*), reflecting some reliance of strain 3D7 on sialic acid for efficient invasion. In the presence of increasing concentrations of anti-CD55 antibody, we observed a dose-dependent inhibition of parasite growth in the NM-treated cells that is similar to untreated cells (~60% versus~40% at maximum concentration of antibody) (*Figure 2B–C*). The finding that CD55 blockade inhibits *P. falciparum* regardless of the presence of membrane sialic acid suggests that CD55 plays a role in both sialic-acid-dependent and -independent invasion. These results are consistent with previous findings showing that the requirement for CD55 is strain-transcendent, including strains that rely on sialic acid to various degrees (*Egan et al., 2015*).

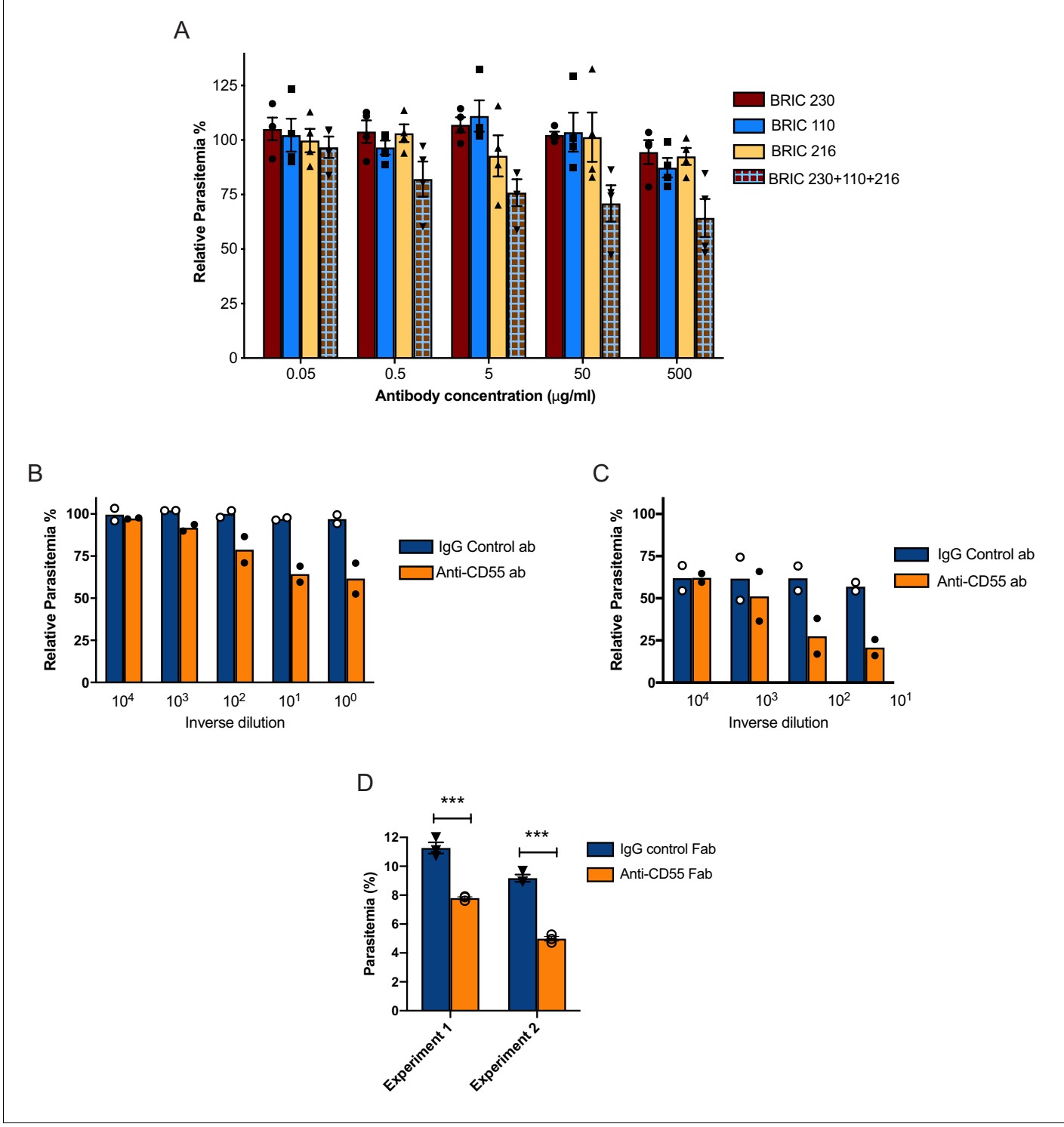

**Figure 2.** Blocking CD55 with antibody inhibits growth of *P. falciparum*. (**A**) *P. falciparum* strain 3D7 parasitemia after 72 hr of growth in RBCS in the presence of increasing concentrations of anti-CD55 monoclonal antibodies, relative to isotype control antibody (BRIC 170) at the same concentration. For the pooled antibodies, the indicated concentration was the total combined value, and there were equimolar amounts of each antibody. (N = 3 biological replicates; n = 2 technical replicates). Error bars indicate SEM. (**B**) Parasitemia of *P. falciparum* strain 3D7 after 72 hr growth in non-enzyme-treated RBCs with increasing concentrations of polyclonal anti-CD55 IgG antibody, relative to that in isotype control antibody at same concentration. (N = 2 biological replicates; n = 2 technical replicates). The highest antibody concentration ($10^0$) was 400 µg/ml. (**C**) As in B, but with neuraminidase-treated RBCs. The highest antibody concentration ($10^1$) was 40 µg/ml. (**D**) *P. falciparum* strain 3D7 parasitemia after 72 hr growth in 400 µg/ml Fab

*Figure 2 continued on next page*

*Figure 2 continued*

fragments generated from anti-CD55 polyclonal antibody or isogenic control. Error bars indicate SEM. \*\*\*p<0.005. The starting parasitemias were 0.3% (Experiment 1) or 0.5% (Experiment 2).

The online version of this article includes the following source data and figure supplement(s) for figure 2:

**Source data 1.** Source data for antibody inhibition assays.
**Figure supplement 1.** Specificity of rabbit polyclonal anti-CD55 antibody.

To confirm that the inhibitory effect of the polyclonal anti-CD55 antibody on *P. falciparum* growth was not due to crosslinking, we tested the growth inhibitory activity of monovalent anti-CD55 antibody fragments (Fab fragments). In the presence of 400 µg/ml of anti-CD55 Fab fragments, we observed ~40% reduction in relative parasitemia, recapitulating the growth inhibitory activity observed with the bivalent anti-CD55 IgG antibody (*Figure 2D*). These results further support the conclusion that CD55 is a critical host receptor for *P. falciparum*.

## Blocking CD55 inhibits *P. falciparum* internalization

To specifically determine the impact of CD55 blockade on *P. falciparum* invasion, we used live cell imaging to visualize and quantify schizont rupture and merozoite invasion in real time in the presence of the anti-CD55 polyclonal antibody or an isotype control (*Figure 3A*; *Videos 1–2*). First, we quantified the efficiency of merozoite internalization in the presence of anti-CD55 antibody versus isotype control. In the presence of the isotype control antibody, of 345 *P. falciparum* merozoites that contacted an RBC, 53 invaded successfully (15%) (*Figure 3B*). This frequency is similar to that described in previous live microscopy studies analyzing the efficiency of 3D7 merozoite invasion in the absence of antibodies (*Volz et al., 2016*; *Weiss et al., 2015*). In comparison, merozoite invasion was significantly reduced in the presence of anti-CD55 antibody. Out of 312 merozoites that made contact with an RBC in the presence of anti-CD55, only 21 successfully invaded (6.7%) (*Figure 3B*). The number of invasion events per schizont rupture was also reduced by half in the presence of anti-CD55 compared to control antibody (*Figure 3C*). In contrast, the efficiency with which egressed merozoites contacted an RBC was similar in the presence of anti-CD55 versus control antibody (>0.5 s; *Figure 3D*). Together, these results indicate that anti-CD55 antibody inhibits entry but does not prevent initial contact between the merozoite and RBC.

## Blocking CD55 has no impact on pre-invasion kinetics

Next, we examined the impact of CD55 blockade on the kinetics of the three main phases of *P. falciparum* invasion: the pre-invasion time (period from initial contact to the onset of internalization), the internalization time, and the time to echinocytosis (transient period of cell dehydration that occurs after internalization) (*Dvorak et al., 1975*; *Weiss et al., 2015*). For the subset of merozoites that ultimately invaded successfully, there was no difference in the length of the pre-invasion time, internalization time, or time to echinocytosis in the presence of anti-CD55 antibody compared isotype control (*Figure 3E–G*). These results indicate that blocking CD55 does not impact pre-invasion kinetics, at least not for the merozoites that manage to invade in the presence of antibody.

## Merozoite-induced erythrocyte deformation is not inhibited by CD55 blockade

During the pre-invasion period, parasite attachment to the RBC is associated with substantial deformation of the host cell membrane as the merozoite reorients apically (*Dvorak et al., 1975*; *Gilson and Crabb, 2009*; *Paul et al., 2015*). The degree of deformation is variable and is mediated by interactions between parasite ligands released from the apical organelles, such as the EBAs and Rhs, and receptors on the RBC membrane (*Weiss et al., 2015*). To investigate whether CD55 plays a role in RBC deformation, we used live microscopy to quantitate the efficiency and kinetics of merozoite-induced RBC deformation in the presence of anti-CD55 antibody versus isotype control. There was a small but significant increase in the efficiency of merozoite-induced deformation in the presence of anti-CD55 antibody compared to the isotype control: approximately 67% of merozoites deformed the RBC in the presence of anti-CD55 antibody, compared to ~60% deformation efficiency with the isotype control (*Figure 4A*). There was no significant difference in the duration of the

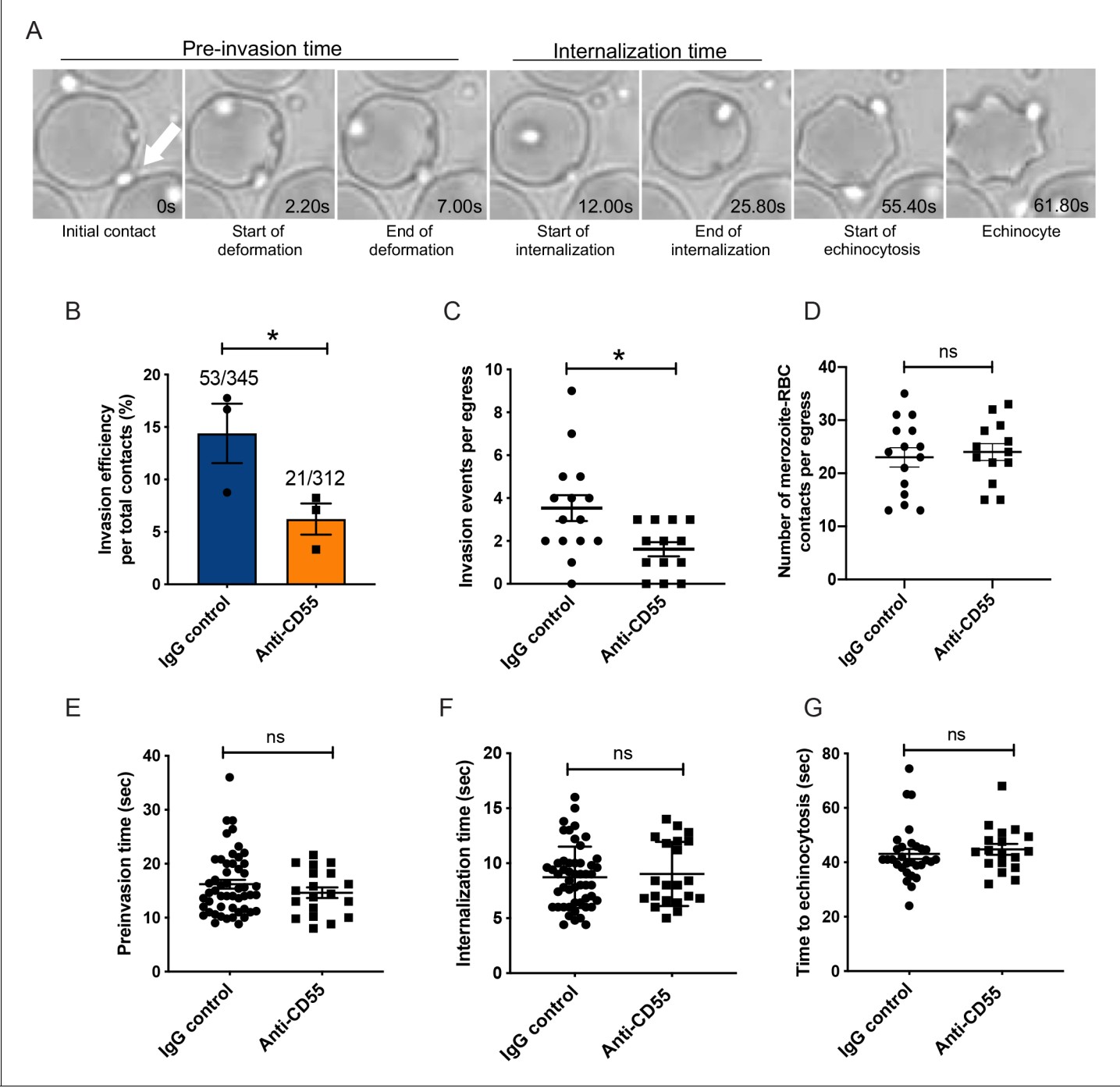

**Figure 3.** Blocking CD55 with antibody inhibits invasion but not pre-invasion kinetics. (**A**) Time-lapse images of invasion after initial merozoite contact. Arrowhead indicates invading merozoite. Time in seconds. (**B**) Percentage of merozoites that invaded an RBC after initial contact in presence of polyclonal anti-CD55 antibody (anti-CD55) or isotype control (IgG control). Bottom number is the total number of merozoites followed that made contact with the RBC, and top number is the subset that invaded. The data were acquired in three independent experiments, and the dots indicate the mean invasion efficiency from each experiment; *p=0.03. (**C**) Number of successful invasion events per schizont rupture (egress) in presence of anti-CD55 antibody or isotype control; *p=0.01. The data were acquired in three independent experiments and the dots indicate individual egress events. (**D**) Number of merozoites that made contact with an RBC (>0.5 s) in the presence of anti-CD55 antibody or isotype control, per egress event. The data were acquired in three independent experiments. (**E**) Pre-invasion time (in s) for merozoites that contact an RBC in the presence of anti-CD55 antibody or isotype control. (**F**) Internalization time (in s) for merozoites in the presence of anti-CD55 antibody or isotype control antibody. (**G**) Time to echinocytosis from the end of merozoite internalization in the presence of anti-CD55 antibody or isotype control. Error bars indicate SEM; ns, not significant (**B– G**).

*Figure 3 continued on next page*

*Figure 3 continued*

The online version of this article includes the following source data for figure 3:

**Source data 1.** Source data for live microscopy studies.

deformation period in the presence of the two different antibodies (*Figure 4B*), suggesting that CD55 does not inhibit nor prolong merozoite-induced deformation.

We employed a four-point deformation scale to quantify the intensity of merozoite-induced RBC deformation in the presence of antibody, where 0 denotes the absence of deformation and three denotes the most extreme degree of deformation (*Weiss et al., 2015*; *Videos 3–5*). Almost all merozoites that invaded successfully induced strong deformation (scores of 2 or 3), regardless of which antibody was present (*Figure 4C*). In comparison, merozoites that failed to invade had lower deformation scores, the majority having scores of 0 or 1. For these merozoites, the distribution of scores skewed higher in the presence of anti-CD55 antibody compared to the isotype control, perhaps reflecting the inability of otherwise 'fit' merozoites to complete invasion (*Figure 4D*). Taken together, these findings demonstrate that blocking CD55 with antibody does not substantially influence the efficiency, duration, or strength of merozoite-induced RBC deformation, and instead suggest that CD55 exerts its effect on invasion at a downstream step.

To further validate the conclusions of the antibody inhibition experiments, we took a complementary approach and performed live cell imaging of *P. falciparum* invasion with natural CD55-null erythrocytes from a rare donor with the Inab phenotype, where CD55 is absent due to a stop codon in exon 2 (*Takahashi et al., 2008*). In traditional invasion assays, we observed ~80% reduction in the efficiency of *P. falciparum* invasion into CD55-null RBCs as compared to control RBCs after ~18 hr, as has been shown previously (*Egan et al., 2015*; *Figure 5—figure supplement 1*). When imaging invasion in real time, we did not observe any successful invasion events into the CD55-null RBCs, out of 310 merozoites that made contact (*Figure 5A* and *Videos 6–7*). In comparison, there were 25 successful invasion events into the control RBCs, out of 197 merozoites that made contact (~11%). Although invasion into the CD55-null cells was clearly impaired, there was no difference in the number of merozoites per egress that contacted an RBC (>0.5 s; *Figure 5B*) nor in the efficiency of merozoite-induced RBC deformation between the two genetic backgrounds (*Figure 5C*). Moreover, we observed no significant difference in the distribution of deformation scores between CD55 null and control RBCs for merozoites that did not invade (*Figure 5D*). These results are consistent with the findings from the antibody inhibition experiments, and suggest that CD55 influences invasion by acting after the period of erythrocyte deformation.

## CD55 acts downstream of rhoptry discharge

Since we observed that the blocking or deletion of CD55 impaired *P. falciparum* entry without altering pre-invasion kinetics or membrane deformation, we hypothesized that CD55 functions after the ligand-receptor interactions that mediate attachment and deformation. During the sequential steps of *P. falciparum* invasion, RBC deformation is followed by injection of the rhoptry organelle contents into the host cell cytoplasm, an event that requires an interaction between the parasite ligand PfRh5 and its RBC receptor, basigin (*Weiss et al., 2015*). The contents of the rhoptries in turn are believed to trigger echinocytosis. When the interaction between PfRH5 and basigin is blocked using inhibitory antibodies and rhoptry discharge is prevented, echinocytosis fails to occur (*Volz et al., 2016*; *Weiss et al., 2015*). To investigate whether CD55 is similarly required for release of the

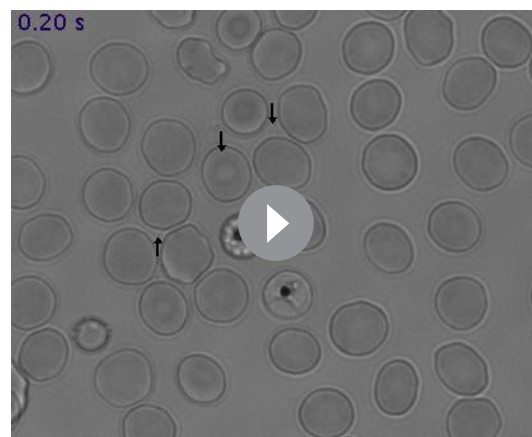

**Video 1.** *P. falciparum* 3D7 invasion in the presence of isotype control antibody.
https://elifesciences.org/articles/61516#video1

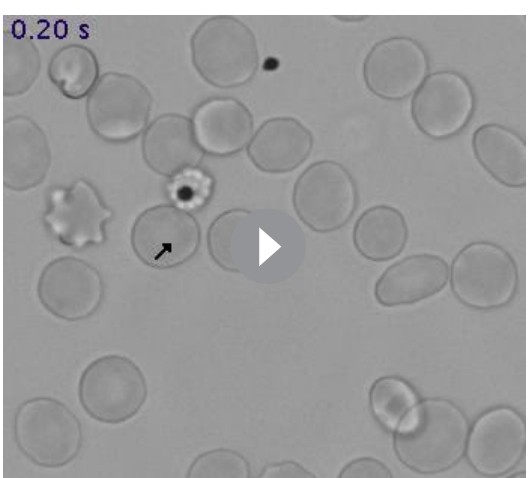

**Video 2.** *P. falciparum* 3D7 invasion in the presence of anti-CD55 antibody.

https://elifesciences.org/articles/61516#video2

rhoptry contents, we quantified the incidence of RBC echinocytosis elicited by attached merozoites in the presence of anti-CD55 antibody versus isotype control using a cytochalasin-D (cyt-D) live microscopy assay (*Figure 6A–B* and *Video 8*). Cyt-D is an inhibitor of actin polymerization that prevents merozoite internalization by inhibiting the actino-myosin motor, but does not impact attachment, rhoptry discharge, or echinocytosis (*Miller et al., 1979*; *Weiss et al., 2015*). Upon treatment with cyt-D,~60% of merozoites that attached to an RBC triggered echinocytosis in the presence of the isotype control antibody, and a similar rate of echinocytosis was observed in the presence of anti-CD55 antibody (*Figure 6C*). These results demonstrate that blocking CD55 does not significantly impact rhoptry discharge or echinocytosis, standing in distinct contrast to what has been observed for basigin and Rh5, where blocking their interaction with antibodies prevents echinocytosis (*Weiss et al., 2015*). Since anti-CD55 antibody does not fully block invasion and might lead to a partial echinocytosis phenotype, we also performed live imaging echinocytosis assays in WT versus CD55-null RBCs to further investigate the impact of CD55 on rhoptry discharge and echinocytosis. The results confirmed that attached merozoites can induce echinocytosis in the absence of CD55, providing additional evidence that CD55 likely acts downstream of rhoptry discharge (*Figure 6D* and *Videos 9–10*). While the efficiency of merozoite-induced echinocytosis stemming from an individual egress event was highly variable in both genetic backgrounds, the overall frequency of echinocytosis was lower in the CD55-null cells compared to WT (17% versus 33%), attributable to several egress events for which echinocytosis was not observed.

Since echinocytosis is an indirect measure of rhoptry discharge, we sought to obtain direct evidence for rhoptry discharge by performing immunofluorescence assays in WT versus CD55-null RBCs to detect the rhoptry bulb marker RAP1. Prior studies of RAP1 have demonstrated that it localizes to the rhoptry bulb as punctate staining in merozoites prior to rhoptry secretion, but after rhoptry secretion it can be detected on the merozoite surface or in/on an attached RBC (*Howard et al., 1984*; *Richard et al., 2009*; *Riglar et al., 2011*; *Schofield et al., 1986*). When free merozoites were allowed to attach to RBCs in the presence of cyt-D, we observed RAP1 staining in a donut or C-shape for merozoites attached to both the WT and CD55-null cells, indicating that rhoptry discharge had occurred (*Figure 6E*). In rare instances, we observed RAP1 staining associated with the RBC, as a 'whorl' in WT cells with an attached merozoite (*Figure 6E*, row 2), and spreading across the RBC surface in CD55-null cells with attached merozoites (*Figure 6E*, row 4). Spreading of RAP1 across the RBC surface has been previously described for merozoites attached in the presence of the R1 peptide, and has been postulated to reflect insufficient sealing of the tight junction (*Riglar et al., 2011*). Together, these findings demonstrate that CD55 acts downstream of the

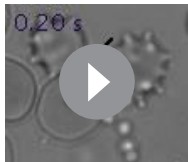

**Video 3.** *P. falciparum* 3D7 merozoite-induced deformation with deformation score 1.

https://elifesciences.org/articles/61516#video3

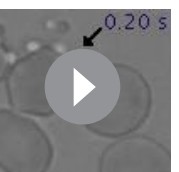

**Video 4.** *P. falciparum* 3D7 merozoite-induced deformation with deformation score 2.

https://elifesciences.org/articles/61516#video4

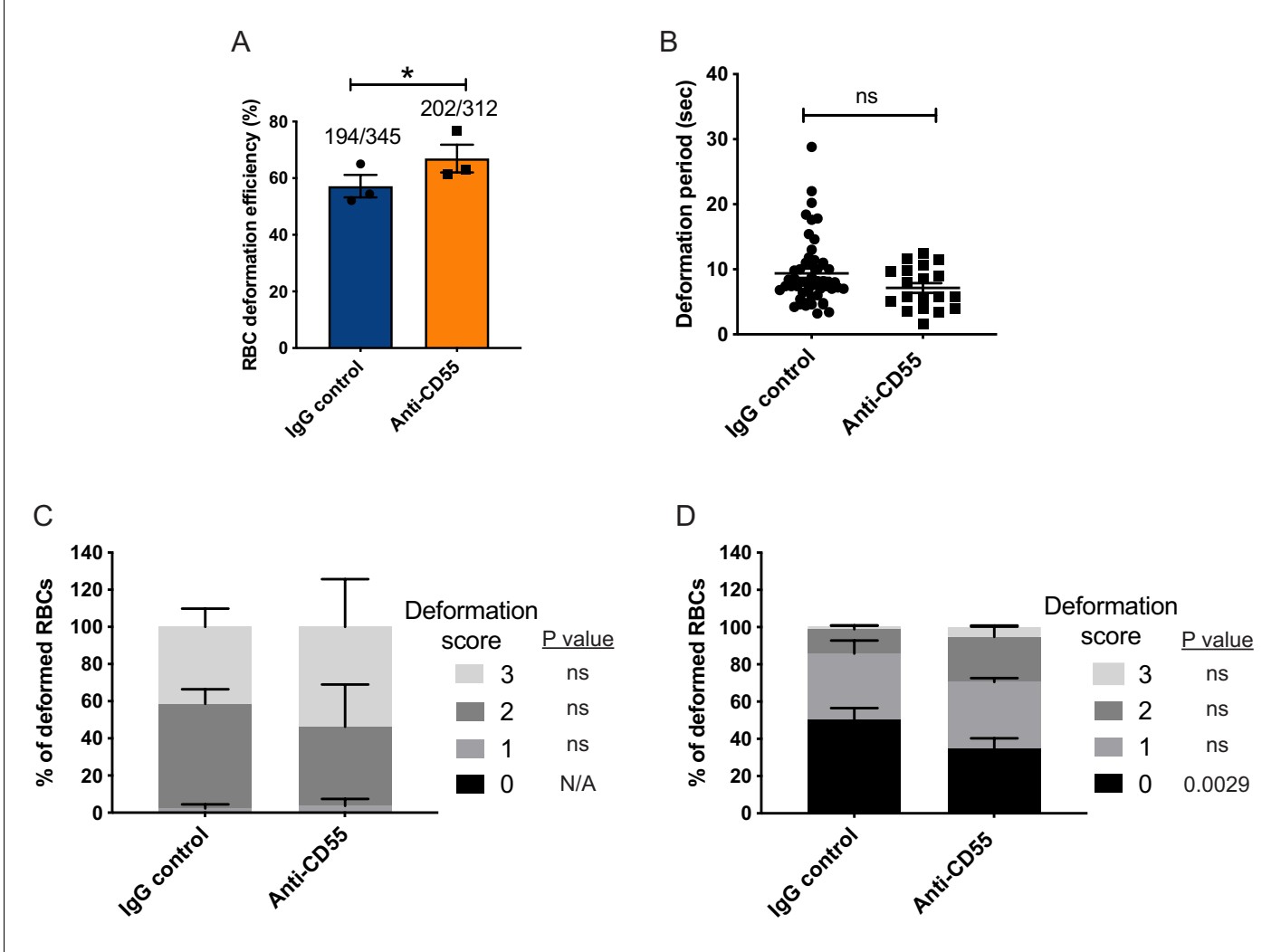

**Figure 4.** Merozoite-induced erythrocyte deformation is not affected by blocking CD55. (A) Efficiency of RBC deformation by merozoites that make contact in presence of anti-CD55 or IgG control. The fractions indicate the number of deformed RBCs out of the total merozoite-RBC contacts observed. The data were acquired in three independent experiments, and the dots represent the mean for each experiment.; *p=0.02. (B) Duration of erythrocyte deformation induced by attached merozoites that ultimately invaded in presence of polyclonal anti-CD55 antibody (anti-CD55) or isotype control (IgG control). Time in seconds. (C–D) Strength of merozoite-induced deformation in presence of anti-CD55 or isotype control for cases where invasion was successful (C) or not successful (D); N/A, not applicable; ns, not significant, *p=0.0029. Error bars indicate SEM (A–D).
The online version of this article includes the following source data for figure 4:

**Source data 1.** Source data for merozoite-induced deformation experiments.

interaction between RH5 and basigin, facilitating a step of invasion that occurs after release of the rhoptry organelles.

## CD55 may be required for progression of the moving junction

The moving junction that forms between the cell membranes of an invading merozoite and the erythrocyte during internalization involves interactions between PfAMA1, which derives from the micronemes, and the PfRON complex, which localizes to the rhoptries. To investigate a possible role for CD55 in formation of the moving junction, we sought to visualize PfAMA1 and PfRON4 as merozoites attempted to invade CD55-null (Inab) or wild-type erythrocytes in the presence of cyt-D. Using a flow cytometry-based attachment assay, we observed a ~ 50% reduction in the number of merozoites attached to CD55-null erythrocytes as compared to WT cells 90 min after addition of late-stage schizonts, as has been observed previously (*Figure 7A*; *Egan et al., 2015*). Using confocal

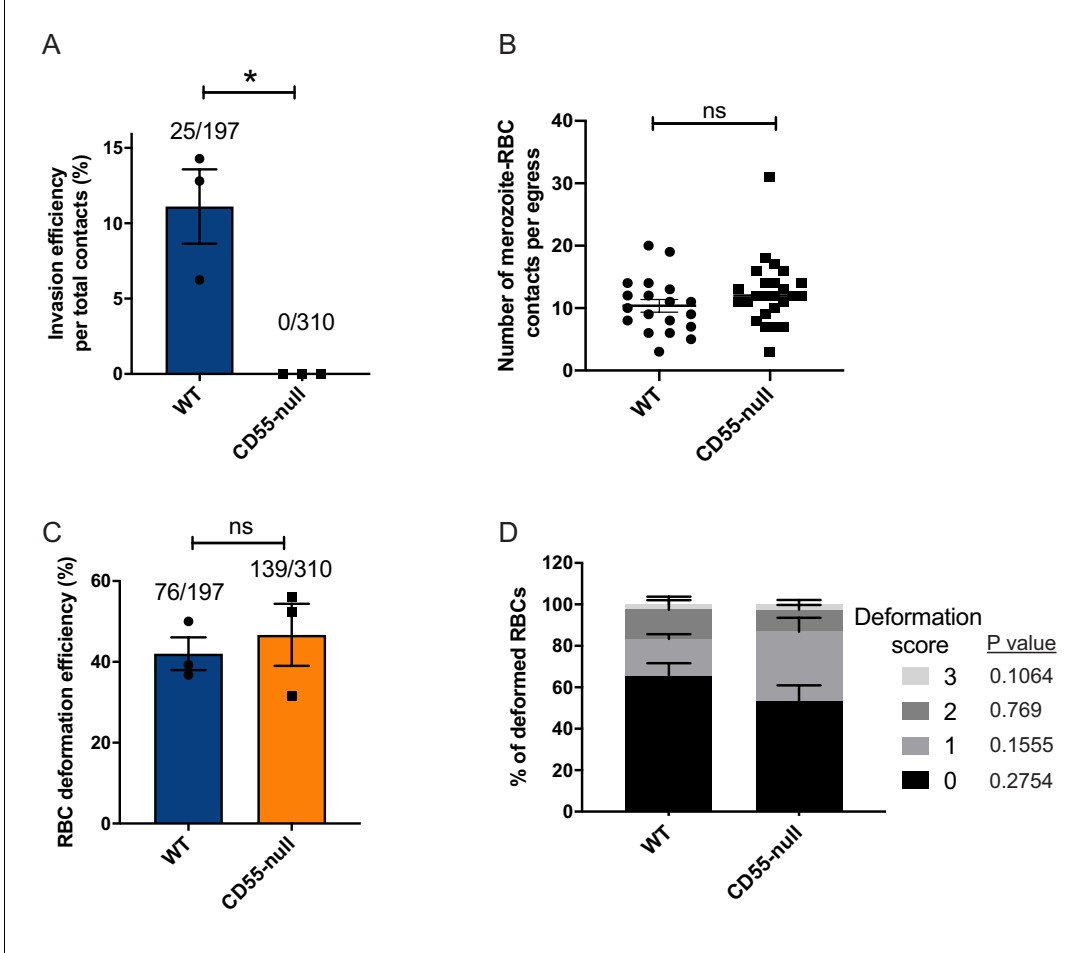

**Figure 5.** Absence of CD55 prevents invasion but does not impact deformation. (**A**) Percentage of merozoites that invaded wild-type (WT) or CD55-null RBCs after initial contact. Bottom number is the total number of merozoites followed that made contact with the RBC, and top number is the subset that invaded. The data were acquired in three independent experiments, and the dots indicate the mean invasion efficiency from each experiment; *p=0.04. (**B**) Number of merozoites per egress that made contact (>0.5 s) with WT or CD55-null RBCs. (**C**) Efficiency of WT or CD55-null RBC deformation upon merozoite contact. Bottom number is the total number of RBCs contacted by a merozoite, and top number is the subset that were deformed upon contact. The data were acquired in three independent experiments, and the dots indicate the mean deformation efficiency for each experiment; ns, not significant. (**D**) Strength of merozoite-induced deformation of WT or CD55-null RBCs among non-invading merozoites; ns, not significant. Error bars indicate SEM (**A– D**).

The online version of this article includes the following source data and figure supplement(s) for figure 5:

**Source data 1.** Source data for live imaging experiments with CD55-null erythrocytes.
**Figure supplement 1.** CD55-null pRBCs are refractory to invasion by *P. falciparum*.
**Figure supplement 1—source data 1.** Source data for CD55-null invasion assays.

microscopy, we found that PfAMA1 and PfRON4 were frequently co-localized at the cellular inter-face for merozoites attached to WT control cells (~85%) (*Figure 7B–C*). In comparison, for mero-zoites attached to CD55-null erythrocytes, PfAMA1 and PfRON4 were co-localized at the cellular interface in only ~40% of cases (*Figure 7B–C*). For those merozoites in which PfAMA1 and PfRON4 were colocalized at the interface, we observed that approximately half of the WT control cells had an indentation at the point of merozoite contact, suggesting the merozoites were mid-invasion, whereas this was never observed for the CD55-null cells (*Figure 7D*). Together, these results support a model where CD55 is required for stability or progression of the moving junction, acting down-stream of junction formation. Consistent with this model, in the flow cytometry-based attachment assay we observed distinct attachment phenotypes for merozoites in the presence of the R1 peptide, which inhibits junction formation, versus merozoites interacting with CD55-null RBCs (*Figure 7E*). In

the presence of R1, we detected almost no merozoites attached to WT cells 90 min after adding late-stage schizonts, whereas attachment to CD55-null cells was only reduced by ~40% at the same timepoint. Together with the immunofluorescence results, these findings suggest that CD55 acts after the moving junction is formed, and instead plays a role in its stability and/or progression.

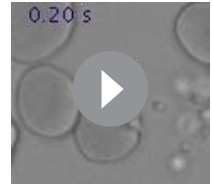

**Video 5.** *P. falciparum* 3D7 merozoite-induced deformation with deformation score 3.
https://elifesciences.org/articles/61516#video5

## Discussion

A comprehensive understanding of the molecular interactions required for *P. falciparum* invasion has been limited by the absence of a robust genetic system to study red cells, which are terminally differentiated and lack a nucleus and DNA. Previously, an RNAi-based forward genetic screen in cultured red cells derived from HSPCs identified two new candidate host factors for invasion: CD44 and CD55 (*Egan et al., 2015*). In this study, we used a combination of genetics and inhibitory antibodies to determine the precise steps of *P. falciparum* invasion during which CD55 functions. We have demonstrated that CD55 is specifically required for merozoite internalization, and plays a unique role relative to the other host receptors known to act during invasion.

Prior work has shown that the efficiency of *P. falciparum* infection is reduced by ~50% in CD55-deficient cRBCs where expression is downregulated using shRNAs. While experiments using erythrocytes from two rare, CD55-null donors suggested that CD55 is essential for *P. falciparum* invasion, this has never been demonstrated genetically. A major roadblock to such experiments has been the technical challenges associated with CRISPR-Cas9 genome editing in primary human hematopoietic stem cells, including cytotoxicity from nucleic acids and low rates of transfection (*Hendel et al., 2015*). In this study, we showed for the first time that fully mature, CD55-null cRBCs can be generated efficiently from primary human HSPCs using CRISPR-Cas9 genome editing. Using isogenic wild-type and mutant cells, we have demonstrated that CD55-null cRBCs are resistant to *P. falciparum* invasion, confirming that CD55 is an essential host factor for *P. falciparum*.

Here, we demonstrated the feasibility and benefits of generating truly mature erythrocytes ex-vivo for the study of malaria host factors, as these cells closely mimic the target cell for *P. falciparum* in the human bloodstream. Our approach involving the co-delivery of two chemically modified sgRNAs together with Cas9 as a ribonucleoprotein complex has been shown to be an effective strategy for CRISPR-based gene knockout in primary HSPCs and T cells (*Hendel et al., 2015*). This method obviates the toxicity associated with plasmid delivery, minimizes off-target activity, and improves sgRNA stability. Combined with our ex vivo erythropoiesis culture system, this method can efficiently generate terminally differentiated, enucleated, CD71-low red cells with a high rate of

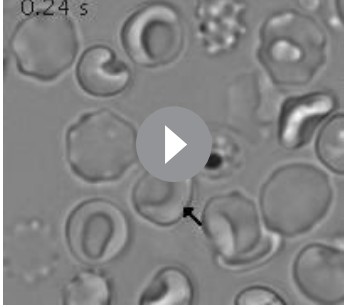

**Video 6.** An example of a failed invasion despite successful merozoite-induced deformation in CD55-null pRBC.
https://elifesciences.org/articles/61516#video6

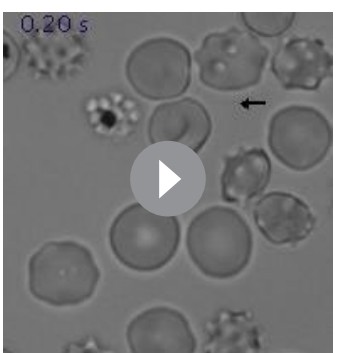

**Video 7.** An example of a successful invasion of a wild-type pRBC.
https://elifesciences.org/articles/61516#video7

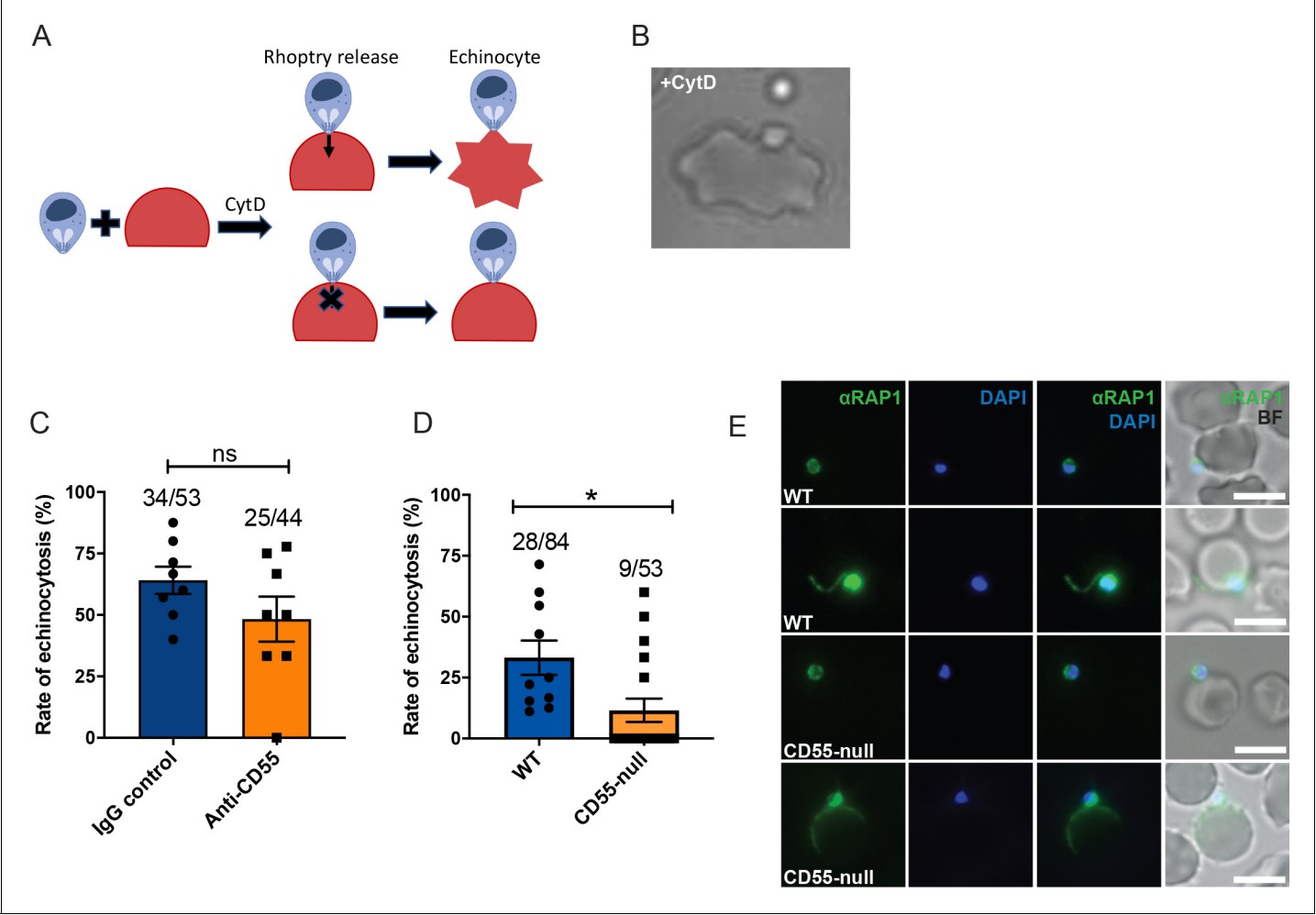

**Figure 6.** Blocking CD55 with antibody does not impact rhoptry discharge. (**A**) Cartoon illustrating echinocytosis elicited by an attached merozoite that has discharged its rhoptry contents in the presence of cyt-D, which prevents internalization. Reagents that block rhoptry discharge prevent echinocytosis. (**B**) Image showing an echinocyte with an attached merozoite. (**C**) Echinocytosis efficiency in the presence of cyt-D and anti-CD55 antibody or isotype control. Bottom number is the total number of merozoite-RBC pairs observed to make sustained contact (>30 s), and top number is the subset in which echinocytosis occurred. Each data point represents the average rate of echinocytosis from a single egress event and error bars represent SEM, obtained in three independent experiments. ns, not significant. (**D**) Echinocytosis efficiency for WT versus CD55-null RBCs with attached merozoites in the presence of cyt-D. Bottom number is the total number of merozoite-RBC pairs observed to make sustained contact (>30 s), and top number is the subset in which echinocytosis occurred. Each data point represents the average rate of echinocytosis from a single egress event and error bars represent SEM, obtained in five independent experiments. *, p<0.05. (**E**) Representative images from immunofluorescence assays showing PfRAP1 localization in WT or CD55-null RBCs with attached merozoites in the presence of cyt-D. Assays were performed twice and ~50 attached merozoites were assessed for each RBC genetic background.

The online version of this article includes the following source data for figure 6:

**Source data 1.** Source data for echinocytosis assays.

gene knock out that can be used to study host genetic determinants for *P. falciparum* in an isogenic background.

We observed that the reliance of *P. falciparum* on CD55 for invasion increased significantly as enucleated cRBCs matured into erythrocytes, likely reflecting the substantial changes in protein abundance that occur during terminal maturation of human red cells, including for proteins known to act as receptors, such as basigin and CR1 (*Gautier et al., 2016*; *Hu et al., 2010*). The efficiency of *P. falciparum* invasion is further influenced by the deformability of the red cell membrane (*Tiffert et al., 2005*), a biophysical property that also changes as red cell progenitors proceed through enucleation and final maturation (*Giarratana et al., 2005*). As clinical malaria is primarily a

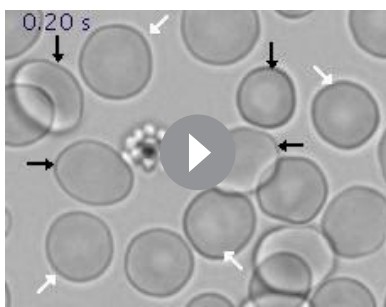

**Video 8.** Echinocytosis induced by attached *P. falciparum* merozoite to pRBCs in the presence of cytochalasin D (cyt-D). Black arrows: Attachment resulting echinocyte formation. White arrows: Attachment failing to result in echinocytosis.
https://elifesciences.org/articles/61516#video8

disease of mature, peripheral blood erythrocytes, our results showing that terminal red cell maturation modifies the requirement for CD55 during invasion highlights the potential drawbacks of using cell lines to study host factors for *P. falciparum*. While considered erythroid in nature, immortalized cell lines such as JK-1, EJ and BEL-A have low rates of enucleation and terminal maturation (*Kanjee et al., 2017*; *Satchwell et al., 2019*; *Scully et al., 2019*), suggesting their utility for genetic experiments on *P. falciparum* invasion may be limited by incomplete phenotypes.

Antibodies targeting a variety of specific host receptors for *P. falciparum* have been shown to have invasion inhibitory activity, including those against GYPA, CR1, and basigin (*Crosnier et al., 2011*; *Pasvol et al., 1989*; *Spadafora et al., 2010*). While we found that individual monoclonal antibodies against three distinct SCR domains of CD55 had no discernable effect on *P. falciparum* growth, in combination they had potent dose-dependent inhibitory activity, suggesting CD55's role in invasion is not restricted to a specific SCR domain. Our demonstration that both the polyclonal anti-CD55 antibody and anti-CD55 Fab fragments can inhibit *P. falciparum* growth aligns with the results from our genetic studies, and corroborates the conclusion that CD55 is a critical host factor for *P. falciparum*. In contrast to CR1, where antibodies are only inhibitory in the absence of red cell sialic acid (*Spadafora et al., 2010*), anti-CD55 antibody blocked both sialic-acid-dependent and -independent invasion. This finding is consistent with prior studies showing that the requirement for CD55 in *P. falciparum* invasion is strain-transcendent (*Egan et al., 2015*), and implies a model where CD55 acts distinctly from CR1 and the other alternative, strain-specific receptors.

Given the inherent limitations associated with studying invasion on a population scale in longer term assays, live microscopy has been increasingly employed to visualize individual *P. falciparum* invasion events in real time. Analysis of the morphology and kinetics of discrete invasion steps in the context of blocking antibodies or soluble proteins has contributed to a model describing the molecular events that occur during invasion (*Volz et al., 2016*; *Weiss et al., 2015*). Our live microscopy experiments have added a new dimension to this model by revealing that merozoite internalization is inhibited in the presence of anti-CD55 antibody or with CD55-null erythrocytes, demonstrating for the first time that CD55 is specifically required for parasite entry. Where and how does CD55 act in relation to the *P. falciparum* ligands and erythrocyte receptors known to function during

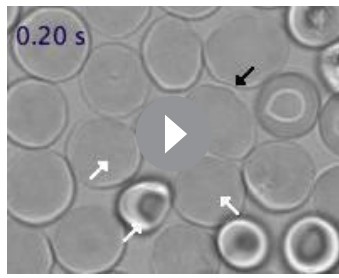

**Video 9.** Echinocytosis induced by attached *P. falciparum* merozoites to previously cryopreserved WT pRBCs in the presence of cyt-D. Black arrows: Attachment resulting in echinocyte formation. White arrows: Attachment failing to result in echinocytosis.
https://elifesciences.org/articles/61516#video9

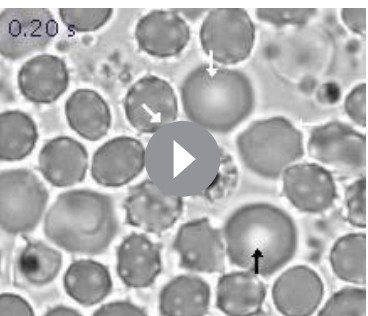

**Video 10.** Echinocytosis induced by attached *P. falciparum* merozoites to previously cryopreserved CD55-null pRBCs in the presence of cyt-D. Black arrows: Attachment resulting in echinocyte formation. White arrows: Attachment failing to result in echinocytosis.
https://elifesciences.org/articles/61516#video10

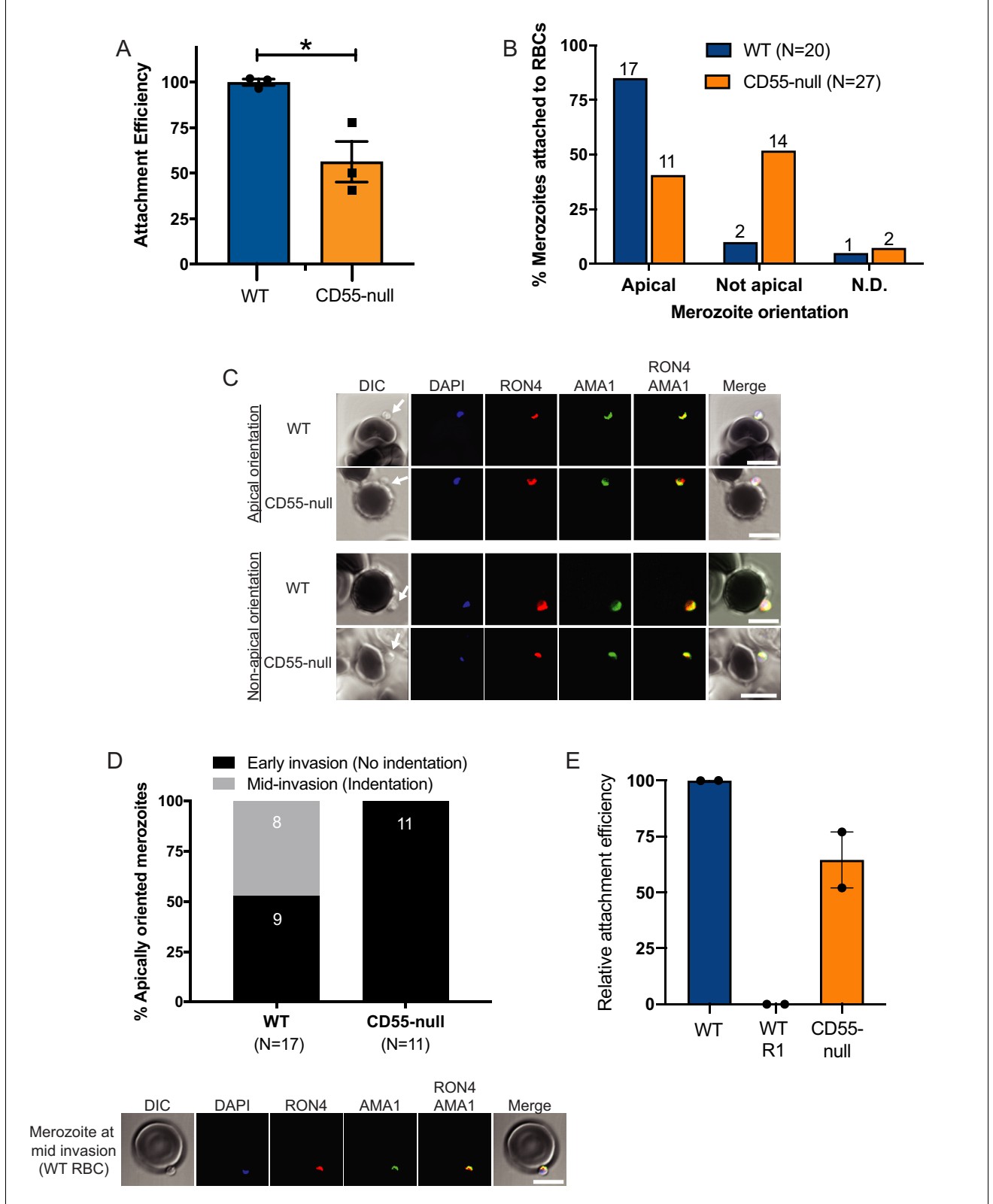

**Figure 7.** Invasion fails to progress in the absence of CD55. (**A**) Attachment efficiency of *P. falciparum* merozoites to CD55-null versus control RBCs in the presence of cyt-D, as measured by flow cytometry 90 min after the addition of synchronized schizonts (N = 3, n = 3). Error bars indicate SEM; *p<0.018. (**B**) Orientation of merozoites attached to WT versus CD55-null RBCs, indicated by the localization of AMA1, RON4, and a density at the merozoite's apical end as measured by confocal microscopy. Apical, AMA1 and RON4 co-localized at the cellular interface and merozoite density

*Figure 7 continued on next page*

*Figure 7 continued*

abutting RBC; Not apical, AMA1 and RON4 not co-localized at the cellular interface and merozoite density not abutting RBC; N.D., indeterminate. The numbers above each bar indicate the subset of attached merozoites in that orientation. The data were acquired in three biological replicates and were scored in a blinded manner. (C) Representative confocal images showing merozoites attached to RBCs with apical or non-apical orientation. Merozoite density at the apical end of the merozoite is indicated by an arrow. (D) Progression of invasion for merozoites apically attached to CD55-null or WT RBCs in the presence of cyt-D. Panel below shows representative confocal images of a merozoite at the mid-invasion stage. (E) Attachment efficiency of *P. falciparum* merozoites to WT RBCs, WT RBCs in the presence of the moving junction-blocking R1 peptide (1 mg/ml), or CD55-null RBCs, all in the presence of cyt-D. Attachment was measured by flow cytometry 90 min after the addition of synchronized schizonts (N = 2, n = 2). Data are presented relative to the efficiency of attachment to WT cells. Error bars represent SEM.

The online version of this article includes the following source data for figure 7:

**Source data 1.** Source data for attachment assays.

invasion? We observed no impact of the anti-CD55 antibody on the number of merozoite contacts with the RBC, pre-invasion kinetics, or merozoite-induced erythrocyte deformability, findings that were validated using CD55-null erythrocytes from a rare donor. These results suggest that CD55 functions distinctly from the 'alternative' receptors required for apical reorientation and red cell deformation (e.g. glycophorins and CR1), as blocking interactions between these receptors and their ligands strongly inhibits merozoite-induced deformability (*Weiss et al., 2015*).

Echinocytosis is a phenomenon of transient red cell shrinkage that commences soon after merozoite internalization. Current evidence suggests that it is stimulated by changes in the red cell cytoplasm that occur once an irreversibly attached merozoite discharges its rhoptry contents into the erythrocyte (*Volz et al., 2016*; *Weiss et al., 2015*). Echinocytosis is inhibited by antibodies that block the interaction between PfRH5 and basigin, implying that this interaction is necessary for rhoptry discharge (*Weiss et al., 2015*). These blocking antibodies also inhibit $Ca^{2+}$ flux into the erythrocyte, possibly reflecting formation of a pore at the cellular interface when PfRH5 and basigin interact (*Volz et al., 2016*). The glycophorins and CR1 act early in the pre-invasion phase during apical reorientation, and blocking their interactions with ligands using genetics, enzyme treatments, or antibody blockade also inhibits echinocytosis (*Volz et al., 2016*; *Weiss et al., 2015*). In contrast, echinocytosis is not prevented by treatment with the actin polymerization inhibitor cyt-D or reagents that block the interaction of PfAMA1 and the RON complex. Together, these findings demonstrate that the stimulus for echinocytosis occurs after the interaction of PfRH5 with basigin, but before establishment of the moving junction.

Our observation that anti-CD55 antibody had no effect on the efficiency of merozoite-induced echinocytosis suggests that CD55 acts after discharge of the parasite's rhoptry organelles. Consistent with this idea, we found that merozoites could stimulate CD55-null RBCs to undergo echinocytosis, which should not happen if rhoptry discharge were prevented. Since the overall rate of echinocytosis was highly variable in the previously cryopreserved WT and CD55-null RBCs used in this study and the anti-CD55 antibody only partially inhibits invasion, we also sought to obtain direct evidence for rhoptry secretion in the absence of CD55. Using immunofluorescence assays, we showed that the rhoptry protein RAP1 localized in a pattern consistent with secreted protein in merozoites attached to both WT and CD55-null RBCs, indicating that rhoptry discharge had occurred. Together, these findings demonstrate that CD55 acts after discharge of the rhoptries, highlighting its role as distinct from basigin and the other established host receptors for *P. falciparum* invasion.

Merozoite internalization requires the formation of a moving junction between the cell membranes of the invading merozoite and the erythrocyte that moves posteriorly down the parasite as it invades. The current model of the moving junction involves interactions between PfAMA1 expressed on the merozoite surface and the PfRON complex inserted into the red cell membrane, independent of any host-encoded receptors (*Alexander et al., 2006*; *Alexander et al., 2005*; *Besteiro et al., 2009*; *Richard et al., 2010*; *Riglar et al., 2011*; *Srinivasan et al., 2011*). In addition to its key role in internalization, the moving junction has also been proposed to be important for formation of the parasitophorous vacuole (PV) (*Yap et al., 2014*).

Could CD55 function as a component of the moving junction during invasion? We observed that PfAMA1 and PfRON4 were almost always co-localized at the cellular interface for merozoites attempting to invade WT cells, whereas this was significantly less common for merozoites attempting to invade CD55-null erythrocytes. For the minority of merozoites that attached to CD55-null

erythrocytes and had PfAMA1 co-localized with RON at the cellular interface, none were observed to have progressed past early invasion, unlike those attached to WT cells. Together, these findings support a model in which CD55 is required for progression of the moving junction, either directly or indirectly. Consistent with this model, irreversible attachment of merozoites to RBCs was completely inhibited in the presence of the junction-blocking R1 peptide, but only moderately inhibited in the absence of CD55, suggesting that CD55 is required not for formation of the moving junction, but for its stability or progression. Additionally, in our immunofluorescence assays of RAP1 localization, we observed a few instances where RAP1 was spread across the RBC surface only in CD55-null RBCs, a phenotype that has been described previously for merozoites attached to RBCs in the presence of R1, and has been speculated to reflect impaired sealing of the moving junction (*Riglar et al., 2011*). Future studies involving higher resolution microscopy such as cryo-EM will be necessary to better characterize how absence of CD55 may impact the moving junction or architecture of the PV after invasion.

Our findings showing that the function of CD55 during invasion can be narrowed to the internalization step are consistent with a model where CD55 acts distinctly from the other known receptors for invasion, potentially by interacting with a specific parasite ligand. As there is a precedent for CD55 on epithelial cells to act as a pathogen receptor that transmits signals to the host cell (*Coyne and Bergelson, 2006*), it is tantalizing to hypothesize that this molecule functions similarly in *P. falciparum* invasion of erythrocytes. Ultimately, biochemical identification of parasite and erythrocyte interaction partners of CD55 will yield important additional insights into the molecular function of CD55 during merozoite internalization. Given the complexity of *P. falciparum* invasion and the unique and essential role of CD55 relative to other established receptors, targeting its activity or interaction partners in novel intervention strategies may enhance the effectiveness of future therapies or vaccines for malaria.

## Materials and methods

### *P. falciparum* culture, invasion assays, and growth inhibition assays

*P. falciparum* strain 3D7, a laboratory-adapted strain obtained from the Walter and Eliza Hall Institute (Melbourne, Australia) was routinely grown in human erythrocytes (Stanford Blood Center) at 2% hematocrit in complete RPMI-1640 supplemented with 25 mM HEPES, 50 mg/L hypoxanthine, 2.42 mM sodium bicarbonate and 0.5% Albumax (Invitrogen) at 37°C in 5% $CO_2$ and 1% $O_2$.

Parasite invasion assays were performed using synchronized late-stage schizont parasites isolated using a MACS magnet (Miltenyi) and added at 1.0–1.5% initial parasitemia to the cultured red blood cells (cRBCs) or peripheral red blood cells (pRBCs) at 0.3% hematocrit. For some invasion assays, cRBCs were enriched for the CD71 negative population by using anti-CD71 antibody-immobilized magnetic beads (Miltenyi), as described below. The CD55-null and WT cRBCs were generated independently for each biological replicate from de-identified isogenic CD34 +HSPCs by RNP transfection (see below). Assays were performed in a volume of 100 µl per well in a 96 well plate, or at 50 µl per well in Half Area 96-well plates (Corning) for the assays with CD71-negative cRBCs. The ring stage parasitemia was determined after 18–24 hr by bright-field microscopy of cytospin preparations stained with May-Grünwald and Giemsa. A minimum of 1000 cells were counted for each technical replicate. For assays in which there was no selection for CD71-negative cells, the invasion efficiency was determined by normalizing the average ring stage parasitema in each genetic background to the average ring stage parasitema in control peripheral blood RBCs (pRBCs) for each biological replicate. For assays that used CD71-negative cRBCs, the ring stage parasitema for each genetic background was normalized to the mean for the control cRBCs. Assays were performed at least three times using two to three technical replicates.

Antibody inhibition assays were performed using pRBCs in the presence of mouse monoclonal antibodies obtained from IBGRL (BRIC 216, BRIC 230, BRIC 110, BRIC 170), polyclonal anti-CD55 antibody produced by New England Peptide, or isotype control rabbit IgG antibody (Novus). All antibodies were dialyzed overnight in RPMI buffer prior to use. Schizont stage parasites were added to untreated or neuraminidase treated pRBCs at 0.5% hematocrit at an initial parasitemia of 0.5% in the presence of 0.05 to 500 µg/ml of the antibodies. For the neuraminidase treatment, the cells were incubated with 66.7mU/ml of neuraminidase from *Vibrio cholerae* (Sigma) at 37°C for 1 hr with

shaking and washed three times in buffer before use in the growth inhibition assays. Some assays were performed in presence of Fab fragments of anti-CD55 or isotype control IgG antibodies at the concentration of 400 µg/ml. The Fab fragments were prepared as described below. Assays were performed two to three times with two to three technical replicates in a volume of 100 µl per well in 96 well plates. Parasitemias were determined on day three by staining with SYBR Green one nucleic acid stain (Invitrogen) at 1:2000 dilution in PBS/0.3% BSA for 20 min, followed by flow cytometry analysis on a MACSQuant flow cytometer (Miltenyi).

## Generation of Fab fragments

Polyclonal anti-CD55 antibody and control IgG were digested using the Pierce Fab preparation kit (Thermo Fisher Scientific). The resulting Fab fragments were quantified on a spectrophotometer, concentrated, and buffer exchanged with incomplete RPMI using 30K Amicon Ultra 0.5 ml centrifugal filter (Millipore). For final quantification, the Fab fragments were stained with Coomassie in a 10% SDS-PAGE gel along with known concentrations of the undigested antibodies, and concentrations of the Fab fragments were determined using ImageJ 1.50i (*Schneider et al., 2012*).

## Generation of cultured red blood cells from primary human CD34 +HSPCs

Bone-marrow-derived primary human CD34 +HSPCs (Stem Cell Technologies) were cultured in erythroid differentiation medium (cPIMDM) composed of Iscove Basal Medium (IMDM) (Biochrom) supplemented with 4 mM L-Glutamine (Sigma), 330 µg/ml holo-transferrin (BBI Solutions), 10 µg/ml of recombinant human insulin (Sigma), 2 IU/ml heparin (Affymetrix), $10^{-6}$ M hydrocortisone (Sigma), 100 ng/ml SCF (R and D Systems), 5 ng/ml IL-3 (R and D Systems), 3 IU/ml Epo (Amgen) and 5% plasma (Octapharma) at 37°C in 5% $CO_2$ in air, as previously described (*Egan et al., 2015*; *Giarratana et al., 2011*). On the second day of the culture, the cells were subjected to nucleofection with ribonucleoprotein (RNP) complexes as described below. On day 5, the cells were washed and resuspended in fresh cPIMDM to maintain concentration $\sim 1 \times 10^4$/ml. Between days 8 and 13, the cells were maintained in fresh cPIMDM devoid of IL3 and hydrocortisone at concentration of $<5 \times 10^5$ cells/ml. On day 13, the cells were washed and plated at $7.5 \times 10^5$ - $1.0 \times 10^6$ cells/ml in cPIMDM without IL-3, hydrocortisone, or SCF. On days 15–16, the cells were co-cultured at $1.0 \times 10^6$ cells/ml concentration on a murine stromal cell layer (MS-5) (*Suzuki et al., 1992*), as previously described (*Giarratana et al., 2005*). The cells were harvested on days 18–22 for different experiments. Growth and differentiation were monitored using hemocytometer-based quantification and light microscopy of cytospin preparations stained with May-Grünwald and Giemsa. To quantify the enucleation rate, cRBCs were incubated in Vybrant DyeCycle violet (Life Technologies) (1:10,000) at 37°C for 30 min, followed by flow cytometry analysis on a MACSQuant flow cytometer (Miltenyi).

## CRISPR-Cas9 genetic modification of primary human CD34 + cells

Two sgRNAs targeting human *CD55* exons were designed using the Broad Institute's GPP sgRNA design portal and synthesized as chemically modified sgRNAs by Synthego. CD55-Cr1 is predicted to recognize a sequence in exon 1 of CD55 and has the sequence: GGGCCCCUACUCACCCCACA. CD55-Cr8 is predicted to recognize a sequence in exon two and has the sequence: CUGGGCAUUAGGUACAUCUG. Experiments using dual sgRNAs typically produce large deletions between the Cas9 binding sites as well as smaller indels, leading to frameshift mutations (*Mandal et al., 2014*). Ribonucleoprotein (RNP) complexes containing one or both sgRNAs were prepared by slowly adding 300 pmol of each sgRNA to 150 pmol Cas9 protein in a 10 µl final volume with nuclease-free water and incubating at room temperature for 10 min. On day 2 after thawing CD34 + cells, the RNP complexes were added to $1 \times 10^5$ cells in 40 µl of P3 nucleofection buffer from the 4D-Nucleofector X kit (Lonza). Half of the mixture was loaded to each well of a 16-well nucleofection cassette and nucleofected using the using E0-100 program with the 4D-Nucleofector Lonza Amaxa. After nucleofection, cells were transferred to 6 ml fresh cPIMDM and incubated at 37°C in 5% CO2 in air.

## Enrichment of CD71-negative cRBCs

Fully differentiated cRBCs were washed in degassed and chilled bead buffer (PBS + 0.5% bovine serum albumin +2 mM EDTA), resuspended in the same buffer (45 µl for $6 \times 10^6$ cells), and

incubated the anti-CD71 antibody-immoblized beads (Miltenyi) (10 µl beads for $6 \times 10^6$ cells) at 4°C for 15 min. The cells were washed in 1 ml of bead buffer and passed through an LS magnetic column (Miltenyi) and washed three times with 2 ml of the buffer. The flow through containing CD71- negative cells was collected, washed with cRPMI medium and used for invasion assays.

## Detection of cell surface proteins by flow cytometry

Expression of RBC surface proteins was measured in control or knockout cRBCs by flow cytometry. A total of $1 \times 10^6$ cRBCs were washed two times with PBS/0.3% BSA and incubated with primary monoclonal antibodies or fluorochrome conjugated antibodies at 4°C for 1 hr. Antibodies used: anti-CD55 (BRIC 216-PE, IBGRL; 1:50) and anti-CD71-PE (Miltenyi; 1:20). After incubation, the cells were washed two times in PBS/0.3% BSA, followed by flow cytometry analysis on a MACSQuant flow cytometer (Miltenyi).

## Immunofluorescence assays (IFA)

IFAs were performed as previously described, with some modifications (*Tonkin et al., 2004*). For IFAs of CD55-CRISPR cRBCs infected with *P. falciparum*, cells were fixed in 4.0% paraformaldehyde and 0.0015% glutaraldehyde in PBS for 20 min at room temperature and blocked for 1 hr in 3% BSA/PBS. The cells were incubated with anti-CD55 antibody (BRIC 216-PE from IBGRL) at 1:50 concentration for 1 hr at 4°C. Cells were mounted Fluoromount-G with DAPI mounting medium (Electron Microscopy Services) and the fluorescent images were captured with a 60X objective on a Keyence BZ-X700 fluorescence microscope. For IFAs of merozoites attached to pRBCs using anti-AMA1 and anti-RON4, samples were prepared as in the attachment assays (see below), and 60 µl of the samples were fixed in the fixative containing 4.0% Paraformaldehyde and 0.015% Glutaraldehyde for 20 min, washed twice in PBS, and allowed to settle onto a Poly-L-Lysine coated coverslip (Corning). The samples were then incubated in 0.1% Triton X-100/PBS for 10 min at room temperature, washed in PBS, and incubated in 0.1 mg/ml of NaBH4/PBS for 10 min at room temperature. Following a wash in PBS, the cells were blocked overnight in fresh PBS/3.0% BSA at room temperature. Samples for RAP1 IFAs were prepared in a similar manner, except these assays utilized free merozoites prepared from E64-treated schizonts by syringe lysis (*Boyle et al., 2010*), which were incubated with previously cryopreserved WT or CD55-null RBCs at 37°C for ~10 min. Primary antibodies: mouse monoclonal anti-PfAMA1 1F9 (1:200) (*Coley et al., 2001*), rabbit polyclonal anti-PfRON4 (1:200) (*Richard et al., 2010*), or mouse monoclonal anti-PfRAP1 209.3 (1:500) (*Howell et al., 2005*) were diluted in blocking buffer applied to cells for 2 hr at room temperature. Following three washes, the cells were incubated in corresponding secondary antibodies at 1:500 dilution: Alexa Fluor 555 (goat anti-rabbit IgG) and Alexa Fluor 488 (goat anti-mouse IgG) for 1 hr at room temperature. The cells were washed in PBS supplemented with 0.1 ng/µl of DAPI (Thermofisher) and mounted in Vectashield anti-fade mounting medium. Imaging for RAP1 IFAs was performed using a Keyence BZ-X700 All-in-One Fluorescence microscope and the image analysis was performed with Fiji. Confocal imaging was performed using a Zeiss LSM710 confocal microscope and the image analysis was performed using Fiji/imageJ (Version 2.0.0-rc-69/1.52i). Apical orientation was defined as PfAMA1 and PfRON4 co-localization at the cellular interface, and non-apical orientation was defined as PfAMA1 and PfRON4 not co-localized at the cellular interface. Merozoites were defined as being mid-invasion if AMA1 and RON4 were co-localized at cellular interface and the merozoite appeared to protrude into the RBC, creating an indentation. Scoring for apical versus non-apical orientation and early versus mid-invasion was performed in a blinded manner by two investigators.

## Anti-CD55 polyclonal antibody generation

A CD55 cDNA (Asp 35-Ser 353) was cloned into a modified pTT5 vector (*Raymond et al., 2011*), whose expression cassette consists of eGFP fused to a puromycin resistance gene followed by a 2A skip peptide (*Funston et al., 2008*) and a BiP signal peptide, and the resulting plasmid was verified by sequencing. 1.8L of 293-6E cells grown in Freestyle 293 media (Invitrogen) to $0.8 \times 10e6$ cells/ml were transiently transfected with linear PEI (Polysciences) with 397 ug DNA at a 1:3 DNA:PEI ratio. TN1 tryptone (Organotechnie) was added to a final concentration of 0.5% (w/v) one day after transfection, and cells were grown for an additional 5 days. Cells were spun down and discarded, and the

following reagents were added to the harvested media (final concentration listed): 20 mM Tris pH8, 350 mM NaCl, 5 mM Imidazole pH8, 0.2 mM NiCl2. 6 ml GE IMAC sepharose six beads, and the sample was rocked at 4C for 1 hr. The sample was then poured into a glass column, and washed with 50 ml of TBS (10 mM Tris pH8, 500 mM NaCl) with imidazole at the following concentrations sequentially: 5 mM, 10 mM, 50 mM, and 250 mM. Nickel was stripped from the column with 10 ml 100 mM EDTA. The majority of the CD55-8xHis protein eluted in the 250 mM Imidazole fraction, with a smaller amount of equal purity in the 50 mM imadazole fraction. These were dialyzed separately into PBS pH 7.4 overnight at 4C, concentrated to 10 mg/ml using Amicon Ultra 15 with 30 kDa cutoff (Millipore), snap frozen in liquid nitrogen and stored at −80C until further use. A total of 2.5 mg of CD55-8xHis was used for three immunizations of two New Zealand white-SPF rabbits by New England Peptide. Approximately 50 ml of recovered antiserum was used for negative affinity purification with an 8X HIS column, and subsequently purified with a protein A column. Approximately 15 ml of purified IgG at 1.209 mg/ml concentration was purified and stored at −80C until further use.

## Live cell imaging

Schizont stage *P. falciparum* strain 3D7 parasites at 4–5% parasitemia and 2% hematocrit were tightly synchronized with 2 µM Compound two to prevent schizont rupture (*Collins et al., 2013*). After ~4–6 hr incubation, they were washed three times and allowed to recover in fresh cRPMI for 45–90 min at 37° C. After recovery, anti-CD55 antibody or IgG isotype control was added to the cells at final antibody concentration of 400 µg/ml and 1% hematocrit. In experiments involving previously cryopreserved CD55-null pRBCs (*Takahashi et al., 2008*) or control WT pRBCs, late stage *P. falciparum* schizonts were isolated using a MACS magnet, synchronized with Compound 2, and then mixed with the pRBCs at 4–5% parasitemia and 1% hematocrit. The assays were loaded into a 20 mm diameter Hybriwell sealing system (Grace Bio-labs) customized for live cell imaging. Rupture of the parasite-infected red blood cells and subsequent invasion events were video recorded using a 60X objective Keyence BZ-X700 live microscopy setup at 37°C supplied with 5% $CO_2$ and 1% $O_2$ gaseous environment. Kinetics and morphology of distinct steps of invasion in the video were measured by using ImageJ (ImageJ 1.50i) as described previously (*Schneider et al., 2012*; *Weiss et al., 2015*). Briefly, the following events were quantified in each video: (a) Contacts between merozoites and RBCs that culminated in successful invasion; (b) successful invasion that resulted echinocytosis; (c) contacts between merozoites and RBCs that could not proceed beyond initial contact/deformation; (d) RBCs contacted before successful invasion; (e) period of pre-invasion (initial contact to end of deformation); (f) during of internalization; (g) time to the onset of echinocytosis; (h) efficiency and degree of merozoite-induced deformation of the target RBC using a four-point deformation scale (0,1,2,3); (i) number of egress events. For the live cell imaging of echinocytosis, 1 µM Cytochalasin D was added to inhibit merozoite internalization. Assays were conducted at 1% hematocrit and 4–5% parasitemia as above, except for those comparing echinocytosis in WT versus CD55-null RBCs, in which parasites were synchronized with ML-10 (*Baker et al., 2017*), 35 mm fluorodishes were used rather than Hybriwells, and assays were conducted in room air at 37°C. For each echinocytosis assay video, the following events were quantified: (a) Egress events; (b) merozoite-RBC contacts lasting at least 30 s; (c) merozoite-induced echinocytosis.

## Merozoite attachment assays

Purified schizonts were added at 12.0% parasitemia to previously cryopreserved CD55-null pRBCs (*Takahashi et al., 2008*) or control WT pRBCs at 1.0% hematocrit in the presence of 1 µM Cytochalasin D, 50 U/ml Heparin or none in a final volume of 100 ul per well in a 96 well plate. Experiments assessing the effect of the R1 inhibitory peptide were performed under similar conditions, except synchronized schizonts were added at 20% parasitemia to RBCs at 0.5% hematocrit, and R1 was added at 1 mg/ml (*Riglar et al., 2011*). The mixtures were incubated at 37°C for 90 min and the schizonts were allowed to rupture. To quantify merozoite attachment by flow cytometry, the samples were fixed in 2% Glutaraldehyde and 0.116M sucrose in PBS, washed in PBS/0.3% BSA, and stained with SYBR Green one nucleic acid stain (Invitrogen) at 1:2000 dilution in PBS/0.3% BSA for 20 min, followed by flow cytometry analysis on a MACSQuant flow cytometer (Miltenyi). The assays were performed two to three times with two to three technical replicates. The merozoite attachment rates

to control or CD55-null RBCs were calculated as the percent of RBCs with an attached merozoite, and were normalized for background attachment in the presence of heparin. The merozoite attachment efficiency was calculated for each genetic background in each biological replicate by normalizing to the mean of the attachment rates in WT RBCs.

## Data analysis

Stastistical analyses were performed and the graphs were generated using GraphPad Prism 8 Version 8.0.2 (159) for macOS. Student's t-test was used for statistical analyses throughout the manuscript.

## Acknowledgements

We thank Carrie Lin for technical assistance, and John Boothroyd, Ellen Yeh, Matt Porteus, Manoj Duraisingh, and members of their labs for helpful discussions. We thank Dave Richard, Anthony Holder, and Alan Cowman for antibodies, and Manoj Duraisingh for R1 inhibitory peptide. Compound 2 and ML-10 were kindly provided by Simon Osborne and LifeArc. BS was funded by a T32 training grant in pediatric nonmalignant hematology and stem cell biology (5T32DK098132-05) and ESE was funded through awards from the NIH (DP2HL13718601) and the Stanford Maternal Child Health Research Institute, and is a Tashia and John Morgridge Endowed Faculty Scholar in Pediatric Translational Medicine and a Baxter Faculty Scholar.

## Additional information

### Funding

| Funder | Grant reference number | Author |
| --- | --- | --- |
| NIH Office of the Director | DP2HL13718601 | Elizabeth S Egan |
| National Institute of Diabetes and Digestive and Kidney Diseases | 5T32DK098132-05 | Bikash Shakya |
| Stanford University | | Elizabeth S Egan |
| Donald E. and Delia B. Baxter Foundation | | Elizabeth S Egan |

The funders had no role in study design, data collection and interpretation, or the decision to submit the work for publication.

### Author contributions

Bikash Shakya, Conceptualization, Formal analysis, Funding acquisition, Investigation, Visualization, Methodology, Writing - original draft; Saurabh D Patel, Conceptualization, Resources, Methodology, Writing - review and editing; Yoshihiko Tani, Resources, Methodology; Elizabeth S Egan, Conceptualization, Resources, Formal analysis, Supervision, Funding acquisition, Validation, Investigation, Visualization, Methodology, Writing - original draft, Project administration, Writing - review and editing

### Author ORCIDs

Bikash Shakya (iD) http://orcid.org/0000-0001-9100-0660
Saurabh D Patel (iD) https://orcid.org/0000-0002-5738-3175
Elizabeth S Egan (iD) https://orcid.org/0000-0002-2112-7700

### Decision letter and Author response

Decision letter https://doi.org/10.7554/eLife.61516.sa1
Author response https://doi.org/10.7554/eLife.61516.sa2

## Additional files

### Supplementary files

• Transparent reporting form

### Data availability

All data generated or analyzed during this study are included in the manuscript and supporting files.

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

# Appendix 1

**Appendix 1—key resources table**

| Reagent type (species) or resource | Designation | Source or reference | Identifiers | Additional information |
|---|---|---|---|---|
| Strain, strain background (*Plasmodium falciparum*) | 3D7 | Walter and Eliza Hall Institute, Melbourne, Australia | | |
| Cell line (*Mus musculus*) | Murine stromal cell layer; MS-5 | PMID:1375698 | | Early passage obtained from Luc Douay Lab, Mycoplasma-negative. |
| Biological sample (*Homo sapiens*) | Bone marrow-derived primary human CD34 + HSPCs (de-identified) | Stem Cell Technologies | Catalog numbers 70002.1 or 70002.2 | |
| Biological sample (*Homo sapiens*) | Erythrocytes; RBCs (de-identified) | Stanford Blood Center | | |
| Biological sample (*Homo sapiens*) | CD55-null pRBCs (CROM7) | doi:10.3925/jjtc.54.359; obtained via MTA from Japanese Red Cross Kinki Block Blood Center | | Cryopreserved |
| Biological sample (*Homo sapiens*) | control WT pRBCs | Obtained from Japanese Red Cross Osaka Blood Center | | Cryopreserved |
| Antibody | anti-CD55 (Rabbit polyclonal) | This paper | | Generated at New England Peptide using human CD55 cDNA immunogen (Asp35-Ser353). Negative affinity purified via 8x HIS column and affinity purified by protein A column. WB, FC (75 µg/ml). Growth assay (serial dilutions). Reacts with WT RBCs and not CD55-null RBCs. |
| Antibody | anti-CD55; BRIC 216 (Mouse monoclonal IgG1) | International Blood Group Reference Laboratory, UK | Cat# 9404 | FC (1:3000) Growth assay (serial dilutions) |
| Antibody | anti-CD55; BRIC 230 (Mouse monoclonal IgG1) | International Blood Group Reference Laboratory, UK | Cat# 9428 | FC (1:3000) Growth assay (serial dilutions) |
| Antibody | anti-CD55; BRIC 110 (Mouse monoclonal IgG1); | International Blood Group Reference Laboratory, UK | Cat# 9402 | Growth assay (serial dilutions) |
| Antibody | Anti-Band3 cytoplasmic domain; BRIC 170 (Mouse monoclonal IgG1) | International Blood Group Reference Laboratory, UK | Cat# 9450 | Growth assay (serial dilutions) |
| Antibody | Isotype Control Rabbit IgG antibody (Rabbit polyclonal) | Novus Biologicals | Cat# NB810-56910 | Growth assay (serial dilutions) |
| Antibody | Anti-AMA1 1F9 (mouse monoclonal) | Gift from Alan Cowman PMID:11707616 | | IFA (1:200) |
| Antibody | Anti-RON4 (Rabbit polyclonal) | Gift from Dave Richard PMID:20228060 | | IFA (1:200) Affinity purified |

*Continued on next page*

*Appendix 1—key resources table continued*

| Reagent type (species) or resource | Designation | Source or reference | Identifiers | Additional information |
|---|---|---|---|---|
| Antibody | anti-RAP1/2 209.3 (Mouse monoclonal) | Gift from Tony Holder. PMID:16102004 | | IFA (1:500) |
| Antibody | Anti-CD55-PE; BRIC 216-PE (Mouse monoclonal) | International Blood Group Reference Laboratory, UK | Cat# 9404-PE | FC, IFA (1:50) |
| Antibody | Anti-CD71-PE (Mouse monoclonal) | Miltenyi | Cat # 130-091-728 | FC (1:20) |
| Biological sample (*Homo sapiens*) | Plasma (Octaplas AB pooled solvent/detergent treated human plasma solution) | Octapharma | | |
| Software, algorithm | Two sgRNAs (Guide RNA design) | Designed using GPP web portal Broad Institute | | |
| Software, algorithm | ImageJ | ImageJ 1.50i PMID:22930834 | Wayne Rasband, NIH | |
| Software, algorithm | Fiji | Version 2.0.0-rc-69/1.52i | | |
| Software, algorithm | GraphPad Prism | Version 8.0.2 (159) for macOS | | |
| Sequence-based reagent | CD55-Cr1; sgRNA1 (sgRNA sequence) | This paper | Chemically modified sgRNA | GGGCCCCUAC UCACCCCACA; Targets exon 1 of human CD55; synthesized by Synthego |
| Sequence-based reagent | CD55-Cr8; sgRNA2 (sgRNA sequence) | This paper | Chemically modified sgRNA | CUGGGCAUUAGGUACAUCU; Targets exon 2 of human CD55; synthesized by Synthego |
| Peptide, recombinant protein | Cas9 protein; Cas9 (Cas9-NLS purified protein) | QB3 MacroLab; University of California, Berkeley | | |
| Peptide, recombinant protein | Neuraminidase; Neuraminidase from *Vibrio cholerae* | Sigma | Cat# N7885 | Confirmed to cleave RBC sialic acid through *P. falciparum* invasion assay using sialic-dependent strain W2mef |
| Peptide, recombinant protein | Recombinant human insulin | Sigma | Cat# 91077C | |
| Peptide, recombinant protein | R1 inhibitory peptide; R1 peptide | Gift from Manoj Duraisingh; PMID:29078358 | | |
| Commercial assay or kit | Pierce Fab Preparation kit | Thermo Fisher Scientific | Cat# 44685 | |
| Commercial assay, kit | 4D-Nucleofector X kit | Lonza | Cat# V4XP-3032 | |
| Peptide, recombinant protein | Holotransferrin | BBI solutions | Cat# T101-5 | >98% purity, from human serum |
| Chemical compound, drug | Heparin (Heparin sodium salt) | Affymetrix | Cat# 9041-08-1 | |

*Continued on next page*

*Appendix 1—key resources table continued*

| Reagent type (species) or resource | Designation | Source or reference | Identifiers | Additional information |
|---|---|---|---|---|
| Chemical compound, drug | Hydrocortisone | Sigma | Cat# H2270-100MG | |
| Peptide, recombinant protein | SCF (Recombinant human Stem Cell Factor/c-kit ligand) | R and D Systems | Cat# 255 SC/CF | |
| Peptide, recombinant protein | IL-3 (Recombinant human IL-3 Protein) | R and D Systems | Cat# 203-IL-050 | |
| Peptide, recombinant protein | Epo (Recombinant Epoetin alpha (Procrit) Epo) | Amgen | NDC# 59676-0312-04 | |
| Chemical compound, drug | E64 | Sigma | Cat# E3132-5MG | |
| Chemical compound, drug | Compound 2 | PMID:23675297 | | Gift from Simon Osborne (MTA with LifeArc) |
| Chemical compound, drug | Cyt-D; Cytochalasin-D | Sigma | Cat# C8273-1MG | |
| Chemical compound, drug | ML-10 | PMID:28874661 | | Gift from Simon Osborne (MTA with LifeArc) |
| Other | SYBR Green 1 Nucleic Acid Gel Stain | Invitrogen | Cat# S-7563 | FC: 1:2000 |
| Other | Vybrant DyeCycle Violet | Life Technologies | Cat# V35003 | FC: 1:10,000 |

