## [Decision Letter]

**Acceptance summary:**

During malaria, Plasmodium merozoites invade erythrocytes, a process that relies on the formation of a moving junction through which the parasite actively glides to enter the target cell. Several erythrocyte receptors are known to play key roles during *Plasmodium falciparum* merozoite invasion, all acting upstream of junction formation. More recently, CD55 was identified as an important host factor for erythrocyte infection by *P. falciparum*, yet its precise role was unknown. Here, using CD55-deficient erythrocytes generated by CRISPR-Cas9 and CD55 antibodies, Shakya et al. confirm the important role of CD55 and show that CD55 plays a unique role relative to other known host receptors during *P. falciparum* invasion, possibly for stability and/or progression of the moving junction during internalization. This study provides important novel insights into the process of host cell invasion by *P. falciparum*, and the methodology will be useful for the study of malaria but also in general erythrocyte biology.

**Decision letter after peer review:**

Thank you for submitting your article "Erythrocyte CD55 mediates the internalization of *Plasmodium falciparum* parasites" for consideration by *eLife*. Your article has been reviewed by 3 peer reviewers, one of whom is a member of our Board of Reviewing Editors, and the evaluation has been overseen by Dominique Soldati-Favre as the Senior Editor. The reviewers have opted to remain anonymous.

The reviewers have discussed the reviews with one another and the Reviewing Editor has drafted this decision to help you prepare a revised submission.

Summary:

This is an interesting study that combines CRISPR-Cas9 editing, antibody-based inhibition and live cell imaging to investigate the role of host CD55 during *P. falciparum* invasion of erythrocytes. The targeting of CD55 for invasion by malaria parasites has been established previously and there is a rare CD55-null blood group that was utilized to show the parasites require the protein for invasion. Here, the authors generated CD55-deficient erythrocytes using CRISPR-Cas9 in erythroid precursors, and confirm the important role of CD55 during invasion by *P. falciparum* using erythrocytes differentiated in culture. The authors then show that anti-CD55 antibodies inhibit *P. falciparum* invasion of normal red blood cells, but have no effect on erythrocyte deformation or echinocytosis. Based on these data, the authors conclude that CD55 acts after discharge of the parasite's rhoptries and plays a unique role relative to all other invasion receptors, possibly for progression of the moving junction during internalization. The manuscript is well written and the work well performed. The study provides important novel insights into the process of host cell invasion by *P. falciparum* and is of broad interest, and the developed methodology will be useful for the study of malaria but also in general erythrocyte biology. However, the conclusion that CD55 acts downstream of rhoptry release is only based on indirect evidence, and additional data are required to clarify the role of CD55 relative to other known receptors and during merozoite attachment.

Essential revisions:

1. A crucial question that needs to be addressed is whether CD55 acts upstream of AMA1/RON complex mediated tight junction formation or downstream by preventing the tight junction from moving over the merozoite surface. The conclusion that CD55 acts downstream of rhoptry discharge and junction formation is an important claim and actually the principal novel finding of the study. This conclusion is based mainly on indirect evidence showing that anti-CD55 antibodies do not inhibit echinocytosis, which results from Rh5-Basigin interaction. The authors should provide direct evidence that rhoptry discharge is not altered by antibodies or in the absence of CD55. For example, the authors could use anti-RAP1 antibodies to visualize the progression of invasion/formation of the junction as it is much clearer than AMA1 since secreted RAP1 can be visualized, depending on the orientation, as either a larger "donut" encircling the smaller RON4 donut or a sort of C-shaped form wrapping around the invading merozoite and stopping at the tight junction as defined by the RON4 signal. In cytD treated merozoites, RAP1 can also be seen as whorls inside the host cell. This would help the authors to determine if the tight junction is sealed as if it isn't, the RAP1 signal will not be released inside the RBC but instead will be seen on the merozoite and/or the RBC surface as previously published using the R1 peptide. This would thus provide clear evidence as to whether CD55 is involved at a step upstream of downstream.

In Figure 7A, erythrocyte binding assays indicate that the merozoites do not bind as well to CD55-null erythrocytes as merozoites on WT erythrocytes. To add some precision to this work, an additional two treatments are suggested where WT erythrocytes are treated with anti-basigin IgG and AMA1-blocking R1 peptide. It would be anticipated, that basigin blocked merozoites would attach the most weakly with tight junction blocking R1 peptide binding more strongly. It is expected that if CD55 functions upstream of the AMA1/RON tight junction complex then merozoites should bind similarly to R1 treated WT erythrocytes as to CD55-null erythrocytes. If merozoites bind to CD55-null erythrocytes better than R1 treated merozoites then CD55 may function downstream of the tight junction. The authors may also consider using super-resolution microscopy to document whether merozoites can form a moving junction in CD55-null cells.

2. The role of CD55 in merozoite attachment is not clear. In their previous study, Egan and colleagues reported that CD55-null erythrocytes were refractory to invasion by all isolates of *P. falciparum* because parasites failed to attach properly to the erythrocyte surface. Here, the authors make the same observation with CD55-null erythrocytes (line 415-418). This is in contradiction with the conclusion that CD55 is not involved during the pre-invasion phase, drawn from experiments with antibodies. This discrepancy needs to be addressed. In particular, Figure 3B and 3C show the invasion efficiency per contact or per egress, but it would be useful to directly quantify attachment in these experiments and show the number of contacts established in the absence or presence of the antibody.

Line 291-296, the authors conclude that blocking CD55 does not impact pre-invasion kinetics, at least not for the merozoites that manage to invade in the presence of antibody. Since the antibody only partially inhibits (40-50%), this statement is problematic. What about the subset of merozoites that ultimately did not invade? In fig3D, if CD55 plays a role in internalization, why is there no increase in the length of the pre-invasion time? Along the same line, the AMA1-RON4 localization data (fig7) show that AMA1 and RON4 are less commonly found at the erythrocyte interface. Based on these observations the authors propose that CD55 could play a role for progression of the moving junction. However, the graph in Figure 7B rather suggests a defect in merozoite reorientation, which is not consistent with the main conclusion of the paper.

3. As the anti-CD55 only partially block invasion, it would be more convincing to document that echinocytosis and rhoptry discharge also occur in CD55-null or CD55-knockout cells as efficiently as in WT cells.

[Editors' note: further revisions were suggested prior to acceptance, as described below.]

Thank you for resubmitting your work entitled "Erythrocyte CD55 mediates the internalization of *Plasmodium falciparum* parasites" for further consideration by *eLife*. Your revised article has been evaluated by 3 peer reviewers, one of whom is a member of our Board of Reviewing Editors, and the evaluation has been overseen by Dominique Soldati-Favre as the Senior Editor. The reviewers have opted to remain anonymous. The reviewers have discussed their reviews with one another, and the Reviewing Editor has drafted this to help you prepare a revised submission.

All reviewers agree that this is a very interesting study that is now suitable for publication at *eLife*. However, two reviewers made a few comments that could be addressed – without additional experimental work – to further improve the manuscript (see the individual assessments below).

*Reviewer #1:*

In this revised manuscript, the authors have now addressed all major comments that were previously raised. They included additional data showing that CD55 is not required for rhoptry discharge based on echinocytosis in CD55-null cells (new Figure 6D) and RAP1 IFA (new Figure 6E). They also provide evidence that tight merozoite attachment to RBCs is not abrogated in CD55-deficient RBCs, in contrary to WT cells in the presence of R1 peptide (new Figure 7E), pointing at a possible role downstream of junction formation.

This is a very nice piece of work. However there are a few points that could be addressed in the text to further improve the manuscript.

The RAP1 results point to a defect in sealing of the junction, therefore the authors cannot exclude a role of CD55 in stability of the junction rather than progression. This should be indicated in the abstract (line 22 and line 381 of the marked-up copy: "stability and/or progression of the moving junction").

The new Figure 3D and 5B show initial contacts of >0.5sec. I am not sure whether such a short time frame is really informative. If the authors have the data, it would be more informative to show the proportion of sustained contacts (>30 sec) as done in the echinocytosis experiments (Figure 6C and 6D, methods line 783 of the marked-up copy).

Some quantification could be provided for Figure 6E (RAP1 IFA), at least the number of parasites that were imaged. The RAP1 signal is much more intense in row 2 and 4, where secreted RAP1 is visible, as compared to row 1 and 3 where the signal is weaker and seems to cover the merozoite surface. The authors may check whether these images are representative.

Line 546: long-term attachment (measured in the presence of cytD) does not really occur during a normal invasion process. Maybe "irreversible attachment" would be a more suitable wording.

Line 945 = PfRAP1

*Reviewer #2:*

I think the authors have answered most of the comments in a satisfactory manner.

The combination of the data makes a strong case that CD55 acts after rhoptry discharge. The data also suggest that the sealing of the junction is potentially compromised however, I think caution should be used when interpreting the new data in Figure 6E. In my opinion, the images provided do not show RAP1 staining in a donut or C-shape for merozoites attached to both the WT and CD55- null cells. The staining looks like it is completely surrounding the merozoite. This is what was previously shown to occur when merozoites are incubated with the R1 peptide. The merozoites have likely gone through an abortive invasion i.e reorientation has occurred along with rhoptry discharge however the TJ has not formed properly so instead of RAP1 being confined to the nascent PV, it is found on the merozoite surface or on the RBC, as shown by the authors. Co-staining with RON4 along with 3D reconstruction would be necessary here to demonstrate whether the TJ is sealed or not. This aspect is perhaps not required for this manuscript but it will be interesting to follow up.

That being said, I agree that the new data shows that rhoptry discharge as occurred since RAP1 clear is not inside the merozoite anymore so I think that the manuscript is now suitable for publication pending a slight rewording.

*Reviewer #3:*

This manuscript is ready for publication.

---

## [Author Response]

Essential revisions:1. A crucial question that needs to be addressed is whether CD55 acts upstream of AMA1/RON complex mediated tight junction formation or downstream by preventing the tight junction from moving over the merozoite surface. The conclusion that CD55 acts downstream of rhoptry discharge and junction formation is an important claim and actually the principal novel finding of the study. This conclusion is based mainly on indirect evidence showing that anti-CD55 antibodies do not inhibit echinocytosis, which results from Rh5-Basigin interaction. The authors should provide direct evidence that rhoptry discharge is not altered by antibodies or in the absence of CD55. For example, the authors could use anti-RAP1 antibodies to visualize the progression of invasion/formation of the junction as it is much clearer than AMA1 since secreted RAP1 can be visualized, depending on the orientation, as either a larger "donut" encircling the smaller RON4 donut or a sort of C-shaped form wrapping around the invading merozoite and stopping at the tight junction as defined by the RON4 signal. In cytD treated merozoites, RAP1 can also be seen as whorls inside the host cell. This would help the authors to determine if the tight junction is sealed as if it isn't, the RAP1 signal will not be released inside the RBC but instead will be seen on the merozoite and/or the RBC surface as previously published using the R1 peptide. This would thus provide clear evidence as to whether CD55 is involved at a step upstream of downstream.

We appreciate the reviewers’ suggestion to obtain direct evidence of rhoptry discharge to help support and further characterize our conclusion that CD55 acts downstream of rhoptry discharge. To address this recommendation, we performed immunofluorescence assays of free *P. falciparum* merozoites incubated with WT or CD55-null erythrocytes in the presence of cyt-D to determine localization of the rhoptry protein PfRAP1.

We observed that merozoites attached to RBCs in the presence of cyt-D demonstrated RAP1 staining on the exterior of the merozoite in a donut shape consistent with secreted RAP1, regardless of the genetic background of the RBCs (Figure 6E). These results provide direct evidence that rhoptry discharge occurs in merozoites attached to both WT and CD55-null cells, supporting the main conclusion of the paper that CD55 acts after rhoptry discharge. In the WT RBCs, we identified a few cases where RAP1 staining could be detected inside the cell as a whorl (Figure 6E, 2nd row). In the CD55-null cells, we identified a few cases where RAP1 staining could be seen spreading along the erythrocyte surface (Figure 6E, bottom row), reminiscent of what has been reported for RAP1 localization in the presence of the inhibitory peptide R1 (Riglar et al., 2011). These results suggest that the tight junction may not properly seal in the absence of CD55. Given the rarity of both the whorl and the spreading RAP1 in our data, more work will need to be done to fully understand these phenotypes. However, our results provide additional support for our conclusion that CD55 acts downstream of rhoptry discharge, and we feel that the manuscript is strengthened by inclusion of these data. We have included new text in the results and discussion to describe these findings (Lines 370-387, 531-539, and 562-566).

In Figure 7A, erythrocyte binding assays indicate that the merozoites do not bind as well to CD55-null erythrocytes as merozoites on WT erythrocytes. To add some precision to this work, an additional two treatments are suggested where WT erythrocytes are treated with anti-basigin IgG and AMA1-blocking R1 peptide. It would be anticipated, that basigin blocked merozoites would attach the most weakly with tight junction blocking R1 peptide binding more strongly. It is expected that if CD55 functions upstream of the AMA1/RON tight junction complex then merozoites should bind similarly to R1 treated WT erythrocytes as to CD55-null erythrocytes. If merozoites bind to CD55-null erythrocytes better than R1 treated merozoites then CD55 may function downstream of the tight junction.

While Figure 7A shows an attachment phenotype in the CD55-null cells, it is important to note that these attachment assays measured long-term attachment because they assessed the percentage of RBCs with attached merozoites in the presence of cyt-D ~90 minutes after addition of late-stage schizonts, and thus reflect a mixed population of attached merozoites, some of which egressed up to ~90 minutes prior. Our previously published timecourse of merozoite attachment to WT versus CD55-null cells in the presence of cyt-D showed that an early timepoint after adding schizonts (30 minutes), merozoites are found attached to WT and CD55-null RBCs with similar efficiency, but that over time fewer remain attached to the CD55-null RBCs, leading to the phenotype at 90 min seen in Figure 7A (Egan et al., 2015). These results suggest that CD55 is not required for initial attachment, but rather for long-term (junction-mediated) attachment (see also response to #3). We have revised text in ~ lines 395-398 to clarify the results shown in 7A.

Previous work has shown that the R1 peptide inhibits junction-mediated attachment of merozoites to WT RBCs by preventing the interaction between AMA1 and RON and formation of the moving junction (Riglar et al., 2011). To directly address the question of CD55’s role in long-term attachment relative to formation of the moving junction, we followed the reviewers’ suggestion to quantify merozoite attachment to WT RBCs in the presence or absence of R1 peptide, compared to CD55-null RBCs. Similar to the attachment assays shown in Figure 7A, these assays measured the percent of RBCs with attached merozoites in the presence of cyt-D ~90 minutes after addition of synchronized, mature schizonts, and thus reflect long-term attachment. The results show that in the presence of R1 peptide, almost no RBCs had an attached merozoite at 90 minutes (Figure 7E). In contrast, attachment to the CD55-null RBCs was only reduced by ~ 40% compared to WT RBCs (consistent with Figure 7A and (Egan et al., 2015)). Together, these findings support a model where R1 disrupts formation of the moving junction, leading to a strong phenotype for long-term attachment, whereas CD55 likely acts downstream of junction formation, resulting in a more moderate phenotype for long-term attachment in the CD55-null RBCs. We have added text to the results (Lines 415-422) and discussion (559-562) to describe these findings and how they advance our understanding of the role of CD55 in *P. falciparum* invasion.

The authors may also consider using super-resolution microscopy to document whether merozoites can form a moving junction in CD55-null cells.

We agree that super-resolution microscopy would be a powerful approach to document whether merozoites can form a moving junction in CD55-null cells, and plan to pursue this in future experiments focused on specifically understanding the role of CD55 for stability or progression of the moving junction.

2. The role of CD55 in merozoite attachment is not clear. In their previous study, Egan and colleagues reported that CD55-null erythrocytes were refractory to invasion by all isolates of *P. falciparum* because parasites failed to attach properly to the erythrocyte surface. Here, the authors make the same observation with CD55-null erythrocytes (line 415-418). This is in contradiction with the conclusion that CD55 is not involved during the pre-invasion phase, drawn from experiments with antibodies. This discrepancy needs to be addressed. In particular, Figure 3B and 3C show the invasion efficiency per contact or per egress, but it would be useful to directly quantify attachment in these experiments and show the number of contacts established in the absence or presence of the antibody.

We apologize for the lack of clarity around our understanding of the role of CD55 in merozoite attachment, and have sought to clarify our thinking here and in the manuscript. This issue is also addressed above in our second response.

Attachment of merozoites to RBCs can be viewed as a three-step process: initial contact, apical reorientation, and junction-mediated (long-term) attachment. In our previous work, we reported a time course of attachment to CD55-null or WT RBCs in the presence of cyt-D in order to measure both initial contact and long-term attachment. We observed that initially attachment was similar to WT or CD55-null cells, but over time there was reduced merozoite attachment to the CD55-null RBCs compared to WT RBCs. Since these experiments were performed using a population of late-stage schizonts that rupture over the course of the assay, we interpret this result as indicating that initial attachment is not affected by CD55, but that long-term (junction-mediated) attachment requires CD55. The role of CD55 in long-term attachment is further addressed in our second response above, and in Figure 7.

In the live microscopy experiments reported in this manuscript, we directly observed and quantified contacts between merozoites and RBCs in the presence of anti-CD55 or control antibody for the first time, defined as contact lasting at least three frames (>0.5 seconds). As suggested by the reviewers, in the revised manuscript we have included a figure panel showing the number of merozoite-RBC contacts per egress in the presence or absence of anti-CD55 antibody (new Figure 3D). The results show that the number of merozoite-RBC contacts per egress was similar in the presence of control antibody versus anti-CD55 antibody, indicating that blocking CD55 does not impact the ability of merozoites to attach to RBCs. This has also been added to the text in lines 259-263.

Line 291-296, the authors conclude that blocking CD55 does not impact pre-invasion kinetics, at least not for the merozoites that manage to invade in the presence of antibody. Since the antibody only partially inhibits (40-50%), this statement is problematic. What about the subset of merozoites that ultimately did not invade?

While we can only quantify the pre-invasion time for merozoites that ultimately invade, we feel that it is important to include these data showing that the merozoites that ultimately invaded in the presence of anti-CD55 antibody did not have a prolonged pre-invasion time. For the subset of merozoites that ultimately fail to invade in the presence of anti-CD55 antibody, we show that they do not have any defects in establishing contact with the RBC (Figure 3D), in mediating RBC deformation (Figure 4), nor in eliciting echinocytosis (Figure 6C). We observed similar results for merozoites attempting to invade CD55-null cells (Figure 5 B-D). Together, these findings support our model that CD55 acts during internalization and does not play a role in the pre-invasion period.

In fig3D, if CD55 plays a role in internalization, why is there no increase in the length of the pre-invasion time?

The pre-invasion time encompasses the period between initial contact and the start of internalization (Figure 3A). The data on pre-invasion time shown in Figure 3E only includes merozoites that ultimately invaded, as it is not possible to quantify the pre-invasion time for merozoites that never invade. While we do not know why some merozoites are able to invade in the presence of anti-CD55 antibody, these data show that they do not have an extended pre-invasion time.

Along the same line, the AMA1-RON4 localization data (fig7) show that AMA1 and RON4 are less commonly found at the erythrocyte interface. Based on these observations the authors propose that CD55 could play a role for progression of the moving junction. However, the graph in Figure 7B rather suggests a defect in merozoite reorientation, which is not consistent with the main conclusion of the paper.

In figure 7B we show that in merozoites attached to CD55-null RBCs, AMA1 and RON4 were less commonly co-localized at the cellular interface compared to merozoites attached to WT RBCs. In the context of our live imaging results which show normal levels of merozoite-induced RBC deformation for both CD55-null RBCs and in the presence of anti-CD55 antibody as well as our attachment assay results, we interpret these data to indicate that the moving junction is less stable in CD55-null versus WT RBCs. The new data in Figure 7E help to distinguish between CD55 acting before or after tight junction formation; since long-term attachment was completely inhibited in the presence of R1 but only partially inhibited in CD55-null cells, these data support the model that the interaction between AMA1 and RON is required for tight junction formation, whereas CD55 is likely required for stability or progression of the tight junction. An unstable moving junction in CD55-null cells would be predicted to alter the merozoite’s apical orientation over the long-term. We have enhanced the Discussion section on this topic (Lines 551-569).

3. As the anti-CD55 only partially block invasion, it would be more convincing to document that echinocytosis and rhoptry discharge also occur in CD55-null or CD55-knockout cells as efficiently as in WT cells.

The reviewers raise an important point that it would be more convincing to document that echinocytosis and rhoptry discharge occur in CD55-null cells, instead of focusing solely on the phenotypes in the presence of anti-CD55 antibody. As described in response to Critique 1, in the revision we have included direct evidence of rhoptry discharge for merozoites attached to CD55-null RBCs (Figure 6E). We have also performed live cell imaging echinocytosis assays in WT versus CD55-null erythrocytes, as recommended (Figure 6D). In these experiments, late stage schizonts were incubated with WT or CD55-null erythrocytes in the presence of cyt-D, and the frequency of echinocytosis was quantified for merozoites that attached to an RBC for at least 30 seconds. The results showed that merozoite-induced echinocytosis can occur in CD55-null RBCs, consistent with our model that CD55 acts after rhoptry discharge. However notably there was a wide range of echinocytosis efficiencies per egress for both genetic backgrounds, with low or no echinocytosis more commonly observed in the CD55-null cells. While this finding may indicate a difference in the susceptibility of WT versus CD55-null RBCs to merozoite-induced echinocytosis, the variation largely occurred at the level of egress events (e.g. for some egress events the echinocytosis rate in CD55-null cells was 40-60% whereas for others it was 0%). Therefore our main conclusion from this series of experiments is that CD55-null RBCs can be stimulated to undergo echinocytosis, demonstrating that CD55 is not required for rhoptry discharge (Videos 9 and 10). The RAP1 IFA experiments requested by the reviewers provide further support for this conclusion. These findings are described in the results and Discussion sections (lines 359-387; lines 527-539).

References:

Cowman, A.F., et al. (2017). The Molecular Basis of Erythrocyte Invasion by Malaria Parasites. Cell host and microbe 22, 232-245.

Egan, E.S., et al. (2015). Malaria. A forward genetic screen identifies erythrocyte CD55 as essential for *Plasmodium falciparum* invasion. Science 348, 711-714.

Mandal, P.K., et al. (2014). Efficient ablation of genes in human hematopoietic stem and effector cells using CRISPR/Cas9. Cell stem cell 15, 643-652.

Riglar, D.T., et al. (2011). Super-resolution dissection of coordinated events during malaria parasite invasion of the human erythrocyte. Cell host and microbe 9, 9-20.

[Editors' note: further revisions were suggested prior to acceptance, as described below.]

All reviewers agree that this is a very interesting study that is now suitable for publication at eLife. However, two reviewers made a few comments that could be addressed – without additional experimental work – to further improve the manuscript (see the individual assessments below).Reviewer #1:In this revised manuscript, the authors have now addressed all major comments that were previously raised. They included additional data showing that CD55 is not required for rhoptry discharge based on echinocytosis in CD55-null cells (new Figure 6D) and RAP1 IFA (new Figure 6E). They also provide evidence that tight merozoite attachment to RBCs is not abrogated in CD55-deficient RBCs, in contrary to WT cells in the presence of R1 peptide (new Figure 7E), pointing at a possible role downstream of junction formation.This is a very nice piece of work. However there are a few points that could be addressed in the text to further improve the manuscript.The RAP1 results point to a defect in sealing of the junction, therefore the authors cannot exclude a role of CD55 in stability of the junction rather than progression. This should be indicated in the abstract (line 22 and line 381 of the marked-up copy: "stability and/or progression of the moving junction").

We agree with the reviewer’s point that we cannot exclude a role of CD55 in stability of the junction rather than its progression. We have added this to the abstract (line 21) and text (line 414) as suggested, and it is also indicated on lines 414 and 562.

The new Figure 3D and 5B show initial contacts of >0.5sec. I am not sure whether such a short time frame is really informative. If the authors have the data, it would be more informative to show the proportion of sustained contacts (>30 sec) as done in the echinocytosis experiments (Figure 6C and 6D, methods line 783 of the marked-up copy).

For the echinocytosis experiments, assays were performed in the presence of Cyt-D and we observed many sustained contacts lasting > 30 seconds. However, such sustained contacts were never observed in the experiments that did not use Cyt-D, because by 30 seconds the merozoites had either invaded or detached. Therefore, the time frame used in Figure 3D and 5B was chosen to be able to quantify all contacts beyond those that were incidental.

Some quantification could be provided for Figure 6E (RAP1 IFA), at least the number of parasites that were imaged. The RAP1 signal is much more intense in row 2 and 4, where secreted RAP1 is visible, as compared to row 1 and 3 where the signal is weaker and seems to cover the merozoite surface. The authors may check whether these images are representative.

The results shown in Figure 6E are representative results from two different experiments in which free merozoites were incubated with WT or CD55-null RBCs in the presence of cyt-D and immunofluorescence assays were performed with anti-RAP1 antibody. We have confirmed that the images shown in rows 1 and 3 are representative of the many attached merozoites that we visualized in these two experiments (~ 50 for each genetic background). We have added the number of experiments performed approximate number of merozoites observed to the legend of Figure 6E (lines 958-959). As mentioned in the text, the phenotypes of PfRAP1 spreading or in a whorl were only rarely observed and thus cannot be considered representative (lines 379-384).

Line 546: long-term attachment (measured in the presence of cytD) does not really occur during a normal invasion process. Maybe "irreversible attachment" would be a more suitable wording.

We have changed the wording in line 559 from “long-term attachment” to “irreversible attachment” as recommended.

Line 945 = PfRAP1

We have fixed this typo as recommended.

Reviewer #2:I think the authors have answered most of the comments in a satisfactory manner.The combination of the data makes a strong case that CD55 acts after rhoptry discharge. The data also suggest that the sealing of the junction is potentially compromised however, I think caution should be used when interpreting the new data in Figure 6E. In my opinion, the images provided do not show RAP1 staining in a donut or C-shape for merozoites attached to both the WT and CD55- null cells. The staining looks like it is completely surrounding the merozoite. This is what was previously shown to occur when merozoites are incubated with the R1 peptide. The merozoites have likely gone through an abortive invasion i.e reorientation has occurred along with rhoptry discharge however the TJ has not formed properly so instead of RAP1 being confined to the nascent PV, it is found on the merozoite surface or on the RBC, as shown by the authors. Co-staining with RON4 along with 3D reconstruction would be necessary here to demonstrate whether the TJ is sealed or not. This aspect is perhaps not required for this manuscript but it will be interesting to follow up.

We appreciate the reviewer’s feedback on the data in Figure 6E. We agree that our data are not conclusive with regards to the sealing of the tight junction in the absence of CD55 and in future work we plan to continue pursuing this line of investigation using RON4 antibodies and 3D reconstruction, as suggested.

That being said, I agree that the new data shows that rhoptry discharge as occurred since RAP1 clear is not inside the merozoite anymore so I think that the manuscript is now suitable for publication pending a slight rewording.